# COCO/DAND5 inhibits developmental and pathological ocular angiogenesis

Natalija Popovic[1,2,†] ![iD], Erika Hooker[1,2,†] ![iD], Andrea Barabino[2,3], Anthony Flamier[2,3,‡], Frédéric Provost[2] ![iD], Manuel Buscarlet[2], Gilbert Bernier[1,2,3,*] ![iD] & Bruno Larrivée[1,2,4,**] ![iD]

## Abstract

**Neovascularization contributes to multiple visual disorders including age-related macular degeneration (AMD) and retinopathy of prematurity. Current therapies for treating ocular angiogenesis are centered on the inhibition of vascular endothelial growth factor (VEGF). While clinically effective, some AMD patients are refractory or develop resistance to anti-VEGF therapies and concerns of increased risks of developing geographic atrophy following long-term treatment have been raised. Identification of alternative pathways to inhibit pathological angiogenesis is thus important. We have identified a novel inhibitor of angiogenesis, COCO, a member of the Cerberus-related DAN protein family. We demonstrate that COCO inhibits sprouting, migration and cellular proliferation of cultured endothelial cells. Intravitreal injections of COCO inhibited retinal vascularization during development and in models of retinopathy of prematurity. COCO equally abrogated angiogenesis in models of choroidal neovascularization. Mechanistically, COCO inhibited TGFβ and BMP pathways and altered energy metabolism and redox balance of endothelial cells. Together, these data show that COCO is an inhibitor of retinal and choroidal angiogenesis, possibly representing a therapeutic option for the treatment of neovascular ocular diseases.**

**Keywords** angiogenesis; COCO; energy metabolism; ocular pathologies
**Subject Category** Vascular Biology & Angiogenesis

## Introduction

Ocular neovascular diseases are a major cause of vision loss in the world. Age-related macular degeneration (AMD) is the third cause of

blindness globally, but it is the primary cause in industrialized countries. Wet AMD arises from the abnormal growth of leaky blood vessels in the subretinal space, which disrupts the function of the heterogeneous cell populations that make up the retina, leading to a partial or complete loss of vision (Ambati & Fowler, 2012; Mitchell *et al*, 2018). While multiple signaling events contribute to the development and progression of pathological neovascularization, vascular endothelial growth factor (VEGF) has long been established as a primary driver of neovascular growth and angiogenesis (Ferrara, 2016; Apte *et al*, 2019). Treatments for neovascular diseases reflect the important role of VEGF in these pathologies. Multiple FDA-approved treatments that target VEGF signaling, including Lucentis, Eylea, and Macugen, have been developed and are in clinical use for the treatment of pathological neovascularization in the eye (Amadio *et al*, 2016). These agents have provided significant clinical benefits to patients afflicted with wet AMD, largely supplanting photodynamic therapy. However, despite great benefits, clinical studies have shown that not all patients respond to anti-VEGF treatments, which could be accounted by the fact that choroidal neovascularization (CNV) associated with AMD is a multifactorial condition whose pathogenesis involves inflammation, angiogenesis, and fibrosis (Kieran *et al*, 2012; Yang *et al*, 2016). Furthermore, all available anti-angiogenic monotherapies are directed specifically to VEGF, which is one of many pathways involved in neovascularization.

Although initially thought to be endothelial-specific, VEGF has been shown to target a variety of non-vascular cells such as neural stem cells, ependymal cells, and neurons including photoreceptors. Indeed, VEGF has been shown to have significant neurotrophic effects by protecting neurons from trauma or disease (Lange *et al*, 2016), although its effects on retinal function are still being debated. Studies using long-term delivery of VEGF inhibitors reported no adverse effects on photoreceptors and retinal function (Ueno *et al*, 2008; Miki *et al*, 2010). However, other studies have reported that VEGF has a survival role on Müller cells and photoreceptors, and that chronic depletion of VEGF results in photoreceptor loss and impaired retinal function (Saint-Geniez *et al*, 2008, 2009).

1  Faculty of Medicine, University of Montreal, Montreal, QC, Canada
2  Hôpital Maisonneuve Rosemont Research Centre, Montreal, QC, Canada
3  Department of Neurosciences, University of Montreal, Montreal, QC, Canada
4  Department of Ophthalmology, University of Montreal, Montreal, QC, Canada
  *Corresponding author. Tel: +1 514 252 3400 4648; E-mail: gbernier.hmr@ssss.gouv.qc.ca
  **Corresponding author. Tel: +1 514 252 3400 7749; E-mail: bruno.larrivee@umontreal.ca
  †These authors contributed equally to this work
  ‡Present address: Whitehead Institute of Biomedical Research, Cambridge, MA, USA

Furthermore, deletion of *Vegfa* from the retinal pigmented epithelium results in an ablation of the choriocapillaris, as well as a loss of cone photoreceptor function (Marneros *et al*, 2005; Kurihara *et al*, 2012). Large multicenter clinical trials, which examined long-term anti-VEGF treatment in patients with AMD, also concluded that therapies that block VEGF could have an effect on the development and progression of geographic atrophy (Martin *et al*, 2012; Grunwald *et al*, 2014). Despite the conflicting evidence in the literature, attention has shifted in recent years to non-VEGF mechanisms of blood vessel formation in the context of providing alternatives to anti-VEGF therapies (Ferrara, 2016).

In addition to VEGF signaling, many other signaling pathways contribute to the development and stabilization of the retinal vasculature. Both canonical and non-canonical Wnt signals have been demonstrated to regulate the retinal vasculature (Zhou *et al*, 2014; Korn *et al*, 2014). Various bone morphogenetic proteins (BMPs) act as both pro- and anti-angiogenic factors (Ntumba *et al*, 2016; Lee *et al*, 2017; Akla *et al*, 2018). TGFβ and Notch signaling pathways are also well-established regulators of angiogenesis and can direct endothelial tip/stalk cell specification and metabolism (De Bock *et al*, 2013). Furthermore, TGFβ has been shown to have pro- and anti-angiogenic activities in neovascular AMD (Tosi *et al*, 2018). COCO (also known as DAND5 or CERL2) is a member of the DAN family (Bell *et al*, 2003). This family is composed of secreted proteins that act as antagonists of soluble BMP, TGFβ, and Wnt molecules (Bell *et al*, 2003). There are seven members of the DAN family: Sclerostin (SOST), uterine sensitization-associated gene-1 (USAG), Gremlin 2 (PRDC; GREM2), Dan (Neuroblastoma Suppressor of Tumorigenicity 1; NBL1), Cerberus (CER1), Gremlin 1 (GREM1), and COCO, which all contain a cysteine-rich DAN domain that is essential for their function (Nolan & Thompson, 2014). The DAN family members have been most widely studied for their roles during development. COCO in particular has been shown to be involved in establishing anterior–posterior patterning in vertebrates (Belo *et al*, 2017). Its inactivation in mice has been reported to lead to multiple laterality and cardiovascular defects and a significant proportion of animals die perinatally (Araujo *et al*, 2014). A recent study has shown that COCO is widely expressed in the retinal photoreceptor layer and that it is a potent inducer of human embryonic stem cell differentiation into cone photoreceptors through inhibition of Activin, BMP, and Wnt signaling (Zhou *et al*, 2015).

As the Wnt, TGFβ, and BMP families have all been implicated in both developmental and pathological retinal angiogenesis and knowing that COCO is expressed in the retina postnatally (Zhou *et al*, 2015), we postulated that COCO may be able to regulate angiogenesis. Furthermore, its positive effects on human photoreceptor development (Zhou *et al*, 2015) suggest that it may be a safe target for the treatment of ocular neovascular diseases. In this study, we report that COCO can inhibit both developmental and pathological angiogenesis in the eye. We further demonstrate that intra-ocular injection of COCO does not result in photoreceptor apoptosis or deleterious effects on the stability of mature blood vessels. Mechanistically, we found that exogenous COCO shows little effect on VEGF signaling but localizes to mitochondria and results in decreased ATP production and induction of reactive oxygen species (ROS) in Human Umbilical Vein Endothelial Cells (HUVECs). Our work identifies a novel inhibitor of retinal and choroidal angiogenesis with potential clinical applications for the treatment of neovascular ocular diseases.

## Results

### COCO inhibits angiogenesis by blocking endothelial cell proliferation and migration

To determine whether COCO affects sprouting angiogenesis in macro- and microvascular endothelial cells, HUVECs and Human Retinal Microvascular Endothelial Cells (HRMECs) were cultured in 3D fibrin gels and tube formation was induced in medium supplemented with 25 ng/ml VEGF-A$_{165}$ (thereafter referred as VEGF), with or without increasing concentrations of COCO for 5 days as previously described (Larrivee *et al*, 2012). Quantification of endothelial tubes revealed a significant decrease in vascular tube area with increasing concentrations of COCO in both HUVECs (Fig 1A and D) and HRMECs (Fig 1B and E). The inhibitory effect of COCO on sprouting angiogenesis was also demonstrated using an *ex vivo* model of choroidal angiogenesis. Briefly, the choroidal tissues from C57BL/6J mice were isolated and cultured in Matrigel (Fig 1C). After treatment with or without COCO for 5 days, vascular sprouting area was evaluated. The sprouting area, normalized to controls, was reproducibly decreased in COCO-treated choroidal explants (Fig 1C and F). Together, these data show that COCO significantly inhibits sprouting angiogenesis in endothelial cells of macro- and microvascular origin as well as in choroidal explants.

We next evaluated the cellular mechanisms underlying the inhibitory effects of COCO on endothelial cell sprouting. Endothelial cell migration plays an essential role in neovascularization, as endothelial tip cells will need to migrate in response to VEGF, and constitutes one of the first steps of the angiogenic response (Gerhardt, 2008). To address the effects of COCO on migration, HUVECs were subjected to a wound healing assay. Briefly, a scratch was performed on a confluent monolayer of HUVECs, and wound closure was evaluated at the time of the scratch and again after 18 h. COCO significantly delayed wound closure compared with control treatment, indicating that COCO can prevent VEGF-induced cell migration (Fig 1G and H). Imaging of cells at the wound edge showed that COCO did not significantly affect polarization of the Golgi apparatus toward the leading edge, suggesting that COCO may not act through Rho GTPase Cdc42 and Rac, which are active at the leading edge of polarized cells and are central to polarity regulation (Raftopoulou & Hall, 2004; Fig EV1A–C).

To understand the possible role of COCO in endothelial cell proliferation, HUVECs cultured in complete medium were treated for 18 h in the presence or absence of recombinant COCO, followed by a 1-h EdU pulse. As shown in Fig 1I and J, HUVECs cultured in the presence of COCO showed reduced EdU incorporation when compared to controls. We also verified that the inhibitory effects of COCO on angiogenesis were not associated with increased apoptosis, as HUVECs cultured for 18 h in the presence of COCO did not show differences in the proportion of apoptotic cells (Fig 1K and L). Taken together, these data reveal that COCO displays anti-angiogenic activity by inhibiting endothelial sprouting, migration, and proliferation without affecting apoptosis.

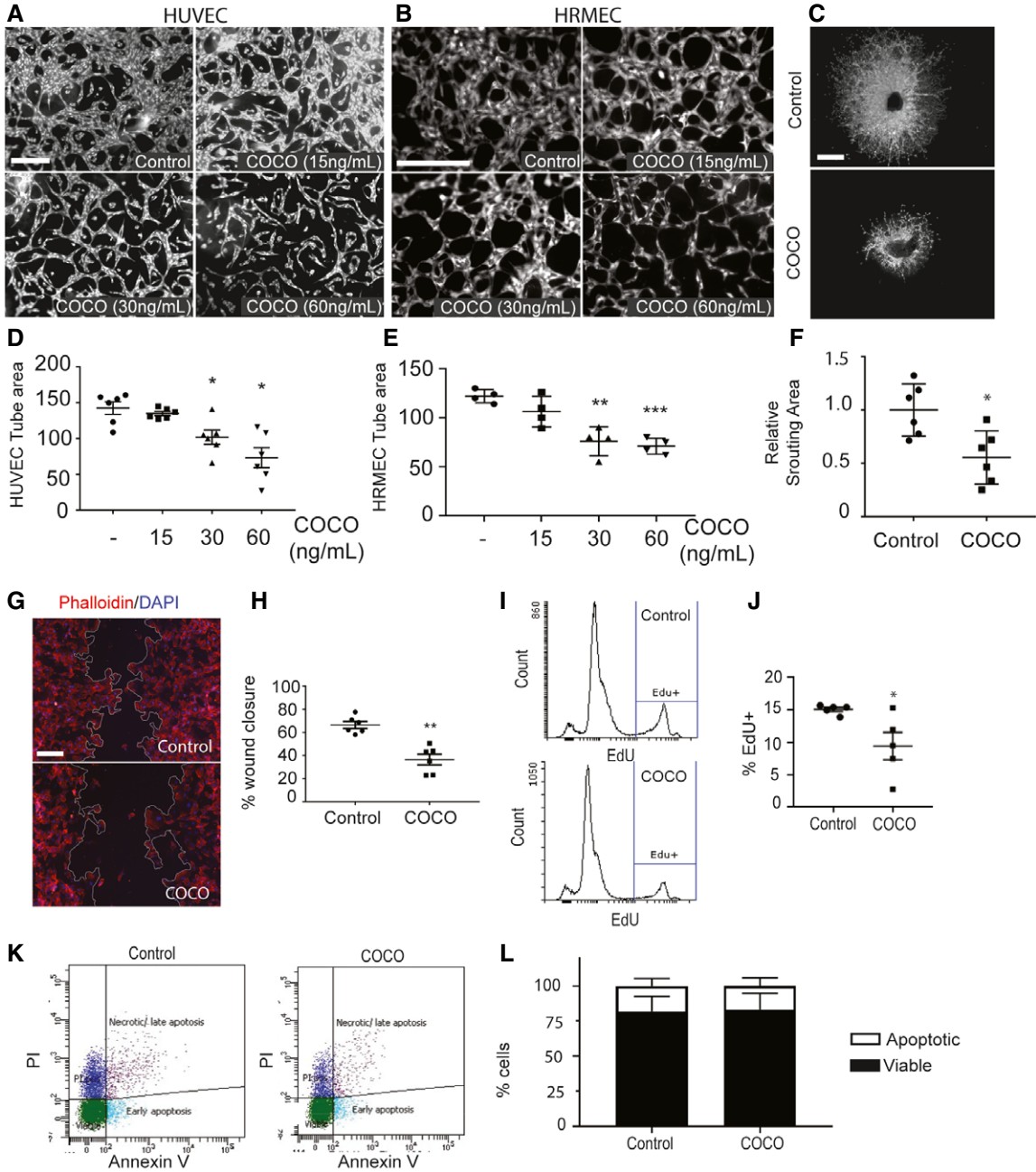

**Figure 1. COCO inhibits sprouting angiogenesis.**

A, B    Representative images of HUVECs (A; *n* = 6) and HRMECs (B; *n* = 4) sprouting in a fibrin gel with VEGF (25 ng/ml) in the presence or absence of COCO. Scale bar, 75 μm.

C    Representative images of choroidal explants cultured for 5 days in the presence or absence of COCO. Scale bar, 500 μm.

D, E    Quantification of tube surface area of micrographs shown in (A, B). Results are presented as mean ± SEM; statistical analyses were performed using Mann–Whitney test. (D: *P = 0.0119 (– vs. COCO 30 ng/ml); *P = 0.0108 (– vs. COCO 60 ng/ml); (E: **P = 0.0013 (– vs. COCO 30 ng/ml); ***P = 0.00026 (– vs. COCO 60 ng/ml)).

F    Quantification of sprouting surface area of micrographs shown in (C; *n* = 6). Results are presented as mean ± SEM, statistical analyses were performed using Mann–Whitney test. *P = 0.0107.

G    COCO decreases scratch wound migration after 18 h. Scale bar, 150 μm.

H    Quantification of wound closure. Results are presented as mean ± SEM, statistical analyses were performed using Mann–Whitney test. **P = 0.0010; (*n* = 6).

I    COCO decreases EdU incorporation in cultured HUVECs.

J    Quantification of EdU incorporation. Results are presented as mean ± SEM, statistical analyses were performed using Mann–Whitney test. *P = 0.0317; (*n* = 5).

K    COCO does not induced apoptosis in HUVECs.

L    Quantification of apoptotic (Annexin V-positive) and viable (Annexin V-negative) cells (*n* = 4).

## COCO inhibits retinal neovascularization

As COCO prevents angiogenic sprouting in cultured endothelial cells, we evaluated whether it could inhibit retinal vascular development. Newborn mouse pups (P1) received intravitreal (ivt) injections of recombinant COCO, and retinas were harvested after 4 days (P5) (Fig 2A). Delivery of exogenous COCO resulted in a significant inhibition of blood vessel development (Fig 2B). Compared with PBS-injected eyes, a pronounced reduction in vessel area (area covered by vessels) and microvessel density (ratio of vessel area to vascularized area) was detected in the retinas of COCO-injected eyes (Fig 2C). The altered vascular pattern was associated with a reduced number of vascular branch points, resulting in a significant reduction in vascular network complexity in COCO-injected retinas. The inhibition of COCO on retinal neovascularization was also as pronounced as that of a VEGF inhibitor (mouse Flt1Fc; R&D systems) (Fig 2B and C). The retinal vasculature of COCO-injected eyes displayed reduced endothelial cell proliferation (Fig EV2A–C) but showed no change in apoptosis (Fig EV2D and E). Blood vessels constrict in the course of vessel regression and endothelial cells retract, leaving behind empty basement membrane sleeves (Korn & Augustin, 2015). However, the retinal vasculature of COCO-treated eyes did not show differences in the number of empty type IV collagen (CollIV) basement membrane sleeves (Fig 2D and E), suggesting that COCO does not affect the switch between vessel maintenance and regression. In spite of its effects on endothelial cells and retinal vascular outgrowth, COCO injections also did not affect retinal pericyte coverage (Fig 2F and G) or photoreceptor apoptosis (Fig 2H and I). Together, these data show that exogenous COCO impairs developmental angiogenesis in the retina and is associated with reduced endothelial cell proliferation.

## Long-term delivery of COCO does not adversely affect photoreceptors

Patients affected by ocular neovascular diseases such as retinopathy of prematurity and wet AMD are typically treated with VEGF inhibitors to control pathological angiogenesis (Amadio et al, 2016). While VEGF inhibitors show good clinical efficacy for the prevention of neovascularization, concerns have been raised since studies have suggested that chronic inhibition of VEGF could adversely affect non-vascular cells in the eye (Nishijima et al, 2007; Lv et al, 2014). We therefore addressed the long-term effects of COCO on photoreceptors and the neural retina. Newborn pups (P1) received weekly injections of COCO or Flt1Fc for 4 weeks (Fig 3A). While there was a mild decrease in the density of retinal vessels in Flt1Fc-injected animals, a striking reduction in blood vessel formation was observed in the retinas of mice that received COCO (Fig 3B and C). Likewise, the number of photoreceptor nuclei, which did not display visible pyknosis, and the thickness of the outer nuclear layer were unaffected in COCO-injected eyes (Fig 3D and E). Thus, even though COCO strongly suppresses retinal neovascularization, it does not appear to compromise photoreceptor survival.

## COCO delivery does not affect mature established vessels in the retina

It is noteworthy that COCO had limited effect on mature retinal vessels, suggesting that COCO may block angiogenesis only in the presence of pro-angiogenic stimuli. This was confirmed in vivo by the observation that COCO inhibition did not affect mature established vessels, when injected in 8-week-old mice for 5 days (Fig 4A). As opposed to newborn pups, which undergo retinal vascular development, a five-day treatment in adult mice showed no difference in the retinal vasculature between PBS and COCO-treated eyes (Fig 4B and C), indicating that COCO mediates its effects by preventing the growth of newly formed vessels, rather than inducing the regression of pre-existing vessels, which is consistent with the absence of changes in empty collagen sleeves (Fig 2D). We extended these findings by injecting adult mice with COCO over a one month-period (Fig 4D). As opposed to newborn pups, which displayed an important reduction in retinal blood vessels outgrowth and density after COCO treatments (Fig 2B and C), no significant effects were observed in adult mice following long-term injections of COCO (Fig 4E and F).

## COCO inhibits pathological neovascularization

The effects of COCO on postnatal developmental angiogenesis led us to evaluate its effects on pathological angiogenesis by subjecting mouse pups to oxygen-induced retinopathy (OIR). Briefly, P7 pups were placed in 75% oxygen for 5 days, leading to vaso-obliteration of the retinal vascular plexus (Stahl et al, 2010). At P12, pups were returned to room air, which stimulates retinal angiogenesis, and leads to the formation of a pathological vascular retinal network characterized by neovascular tufts (Fig 5A). Treatment of COCO at P12 significantly reduced pathological neovascularization in retinas harvested at P17. While revascularization and the avascular region of the central part of the retina was not affected by COCO, the amount and size of neovascular tufts were significantly reduced in the eyes injected with COCO compared with PBS treatment (Fig 5B–D). As with developmental retinal angiogenesis, the anti-angiogenic effects of COCO were similar to those observed with Flt1Fc.

The effects of COCO on CNV were also evaluated by subjecting mice to laser-induced CNV, a model which recapitulates the CNV occurring in wet AMD patients (Lambert et al, 2013). Briefly, 8-week-old mice were subjected to laser impact, followed by intravitreal injections of either COCO, Flt1Fc, or PBS, and CNV was detected 14 days later by staining choroid-sclera whole mounts with IsoB4 (blood vessels) and phalloidin (RPE) (Fig 5E). We observed a significant decrease in the area of CNV in mice treated with COCO and Flt1Fc compared with controls (Fig 5F and G). Together, these observations show that the inhibitory effect of COCO on retinal and choroidal neovascularization is similar to that observed with Flt1-Fc treatment.

## COCO does not directly alter VEGF signaling in primary endothelial cells

VEGF is one of the main drivers of angiogenesis, and inhibition of this pathway has been the main target of anti-angiogenic therapies. Gremlin, a member of the Dan family, has previously been shown to directly interact with VEGFR2 and promote its activity (Mitola et al, 2010). We therefore evaluated whether VEGF signaling was modulated following COCO treatment. HUVECs were starved overnight in 1% FBS in the presence or absence of COCO, followed by

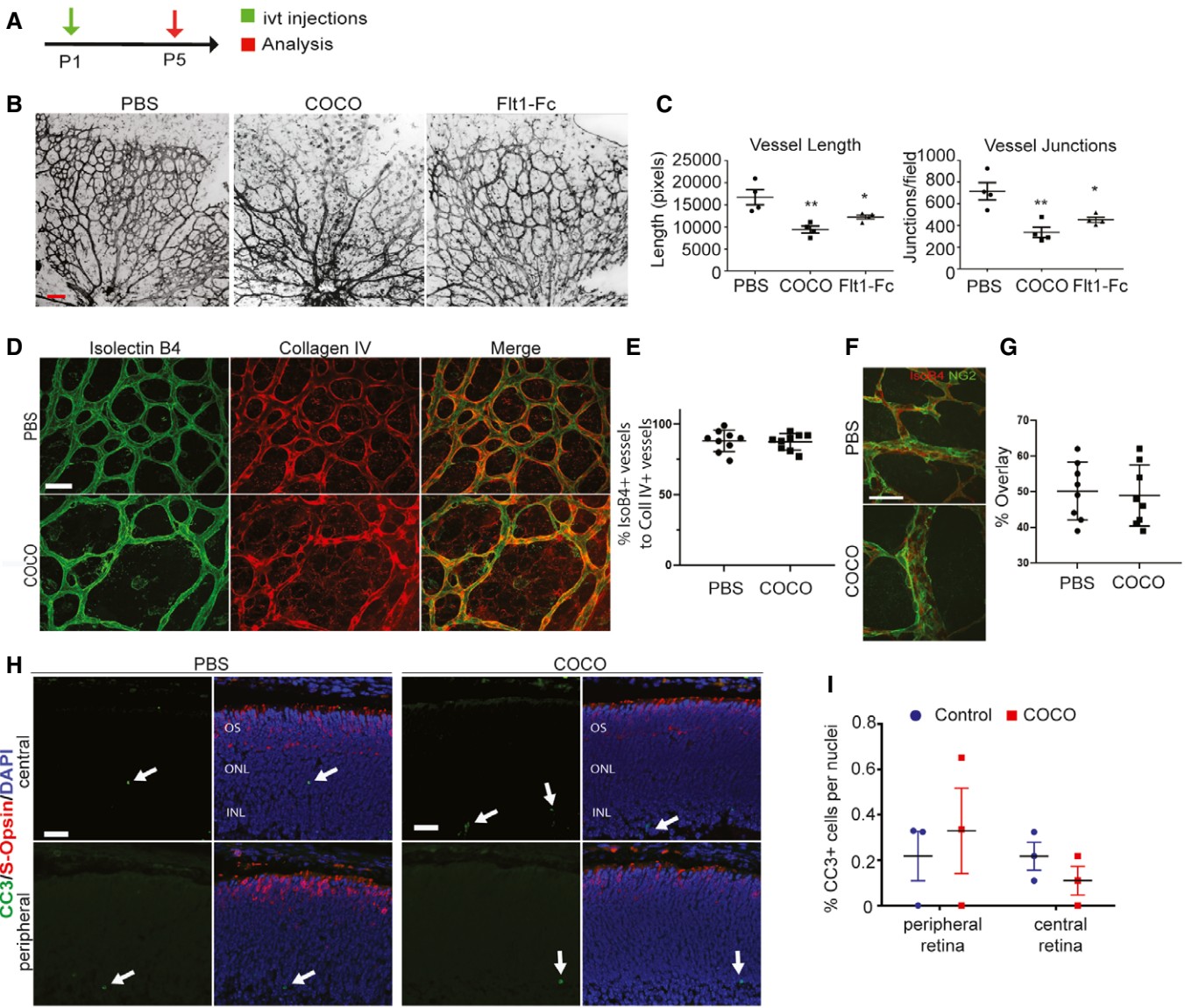

**Figure 2. COCO inhibits developmental retinal angiogenesis.**

A  Schematic of the experimental strategy to assess early formation of the retinal vasculature (P1–P5).
B  Retinal flat mounts of P5 mice injected with PBS, COCO, or Flt1Fc are stained with IB4 (negative images of the fluorescent signal). Scale bar, 100 μm.
C  Quantification of vascular length and number of branchpoints (*P*-values are vs. PBS treatment). Vessel length: **$P = 0.0059$ (PBS vs. COCO); *$P = 0.0169$ (PBS vs. Flt1Fc). Vessel junctions: **$P = 0.0063$ (PBS vs. COCO); *$P = 0.0159$ (PBS vs. Flt1Fc); ($n = 4$ mice/group).
D  COCO does not increase empty collagen IV sleeves. Scale bar, 50 μm.
E  Quantification of % of IsoB4 + vessels to Coll IV + vessels; ($n = 9$).
F  Visualization of pericyte coverage (NG2:green; IsoB4:red) in PBS- or COCO-injected retinas. Scale bar, 50 μm.
G  Quantification of percentage of IsoB4 vascular staining covered by NG2 staining (% overlay) ($n = 8$).
H  COCO injections (P1) do not result in apoptosis in P5 retinas. Scale bar, 40 μm.
I  Quantification of apoptotic cells in control and COCO-injected mice ($n = 3$). Results are presented as mean ± SEM and statistical significance was analyzed by Mann–Whitney test. *$P < 0.05$, **$P < 0.01$.

VEGF stimulation for up to 60 min. Immunoblotting analysis of key signaling regulatory events did not reveal changes in VEGF-induced phosphorylation of VEGFR downstream signaling molecules (Fig 6A and B). In HUVECs cultured in the presence of COCO, VEGFR2 phosphorylation at Y1175 was not significantly altered, nor was the phosphorylation of Akt, Erk, and p38 in response to VEGF stimulation (Fig 6A and B). The expression of VEGF receptors was also not altered following COCO stimulation for up to 24 h (Fig 6C and D). Taken together, these data suggest that the inhibitory effects of COCO do not appear to be a direct consequence of altered VEGF signaling in endothelial cells, although it remains a possibility that downstream cellular events may similarly be modulated by both

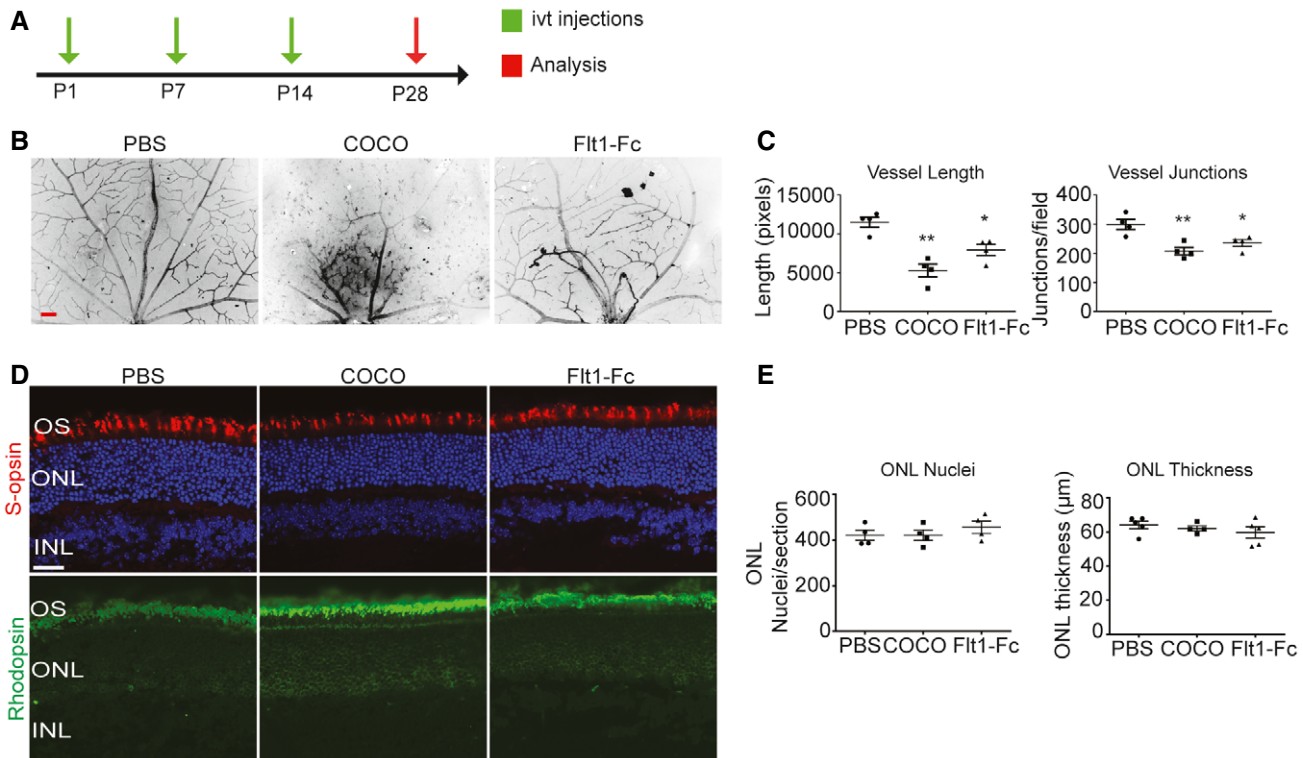

**Figure 3. Chronic injections of COCO decrease blood vessel development but do not result in retinal apoptosis.**

A  Schematic of the experimental strategy to assess early formation of the retinal vasculature (P1–P28).
B  Retinal flat mounts of P28 mice injected with PBS, COCO, or Flt1Fc are stained with IB4 (negative images of the fluorescent signal). Scale bar, 100 μm.
C  Quantification of vascular length and number of branchpoints (P-values are vs. PBS treatment). Vessel length: **P = 0.0019 (PBS vs. COCO); *P = 0.0128 (PBS vs. Flt1Fc). Vessel junctions: **P = 0.0073 (PBS vs. COCO); *P = 0.0302 (PBS vs. Flt1Fc); (n = 4).
D  COCO injections (P1, P7, and P14) do not result in ONL thinning in P28 retinas. Scale bar, 40 μm.
E  Quantification of apoptotic cells (cleaved caspase-3; CC3) in control and COCO-injected mice (P-values are vs. PBS treatment). (n = 4). Results are presented as mean ± SEM and statistical significance was analyzed by Mann–Whitney test. *P < 0.05, **P < 0.01, ***P < 0.001.

factors. We also evaluated whether COCO could potentiate the anti-angiogenic effects associated with VEGF inhibition. *In vitro*, we show that treatment with a sub-optimal concentration of COCO (50 ng/ml) can potentialize the anti-angiogenic effects of Flt1Fc in a sprouting experiment (Fig EV3A and B), although no such potentiation was observed at higher COCO concentrations (75–300 ng/ml COCO), suggesting that the concentration window required to observe an additive effect between COCO and a VEGF inhibitor is relatively narrow. *In vivo*, combination of COCO (50 ng/ml) with Flt1Fc did not show an additive effect on ocular neovascularization (Fig EV3C and D). This lack of additive effects between VEGF and COCO on retinal vascular development may reflect that COCO and VEGF still share common downstream cellular events in endothelial cells that are limiting or tightly regulated. Alternatively, COCO likely also acts on non-vascular cells in the retina, which may indirectly mask the effects of an anti-VEGF on the retinal endothelium.

## Expression of COCO in cultured endothelial cells and in human and mouse retina

To fully understand the multifaceted mechanism of the anti-angiogenic effects of COCO in retinal angiogenesis, we evaluated its endogenous expression in the retina. Western blot analysis of adult human retina tissue showed a 36 kDa band, consistent with the presence of a COCO dimer (Fig 7A; Nolan & Thompson, 2014). COCO was also abundantly detected in human pluripotent stem cell-derived cone photoreceptors (Fig 7B). Immunostaining of adult human retinas revealed the presence of COCO in inner segments of photoreceptors (a mitochondria-rich region), at or close to the outer plexiform layer, and in the ganglion cell layer (Fig 7C). We also confirmed the pattern of COCO expression using the highly sensitive RNAscope *in situ* hybridization technique for visualizing *Dand5* transcripts in mouse retinal sections (Fig 7D). *Dand5* transcripts were detected throughout the neural retina and were also present in the photoreceptor nuclear layer, as shown by co-localization with visual-arrestin immunostaining, in agreement with previous findings (Zhou *et al*, 2015; Fig 7E). In addition, *Dand5* was widely expressed in inner nuclear layer, composed of horizontal, bipolar, and amacrine cells, as well as in the ganglion cell layer. Finally, we next evaluated the expression of COCO in cultured endothelial cells. While immunostaining of unstimulated HUVECs showed no significant COCO immunoreactivity (Fig EV4), HUVECs exposed to recombinant COCO for 5 h displayed COCO immunoreactivity in mitochondria, as shown by co-localization with a mitochondria-

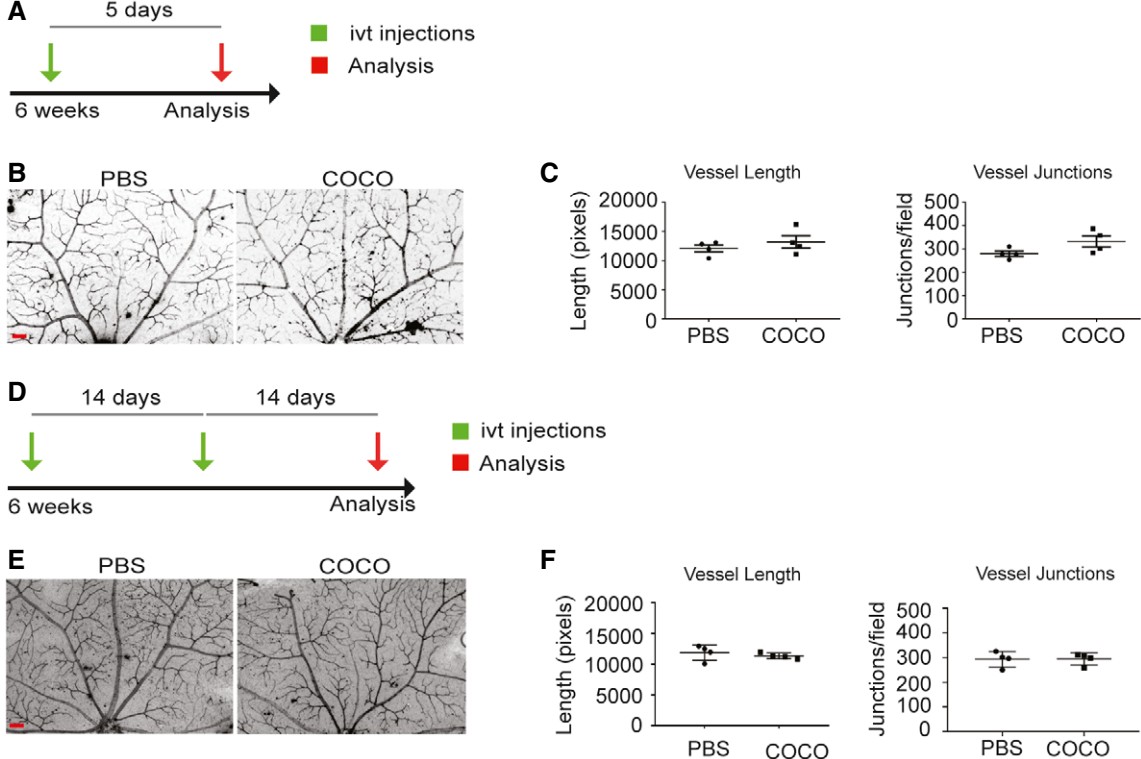

**Figure 4. COCO injections do not induce vascular regression in adult mice.**

A  Schematic of the experimental strategy to assess injections of COCO in the adult retinal vasculature.
B  Retinal flat mounts of adult mice injected with PBS or COCO for 5 days are stained with IB4 (negative images of the fluorescent signal). Scale bar, 100 μm.
C  Quantification of vascular length and number of branchpoints; (n = 4).
D  Schematic of the experimental strategy to assess repeated injections of COCO in the adult retinal vasculature.
E  Retinal flat mounts of adult mice injected with PBS or COCO for 28 days are stained with IB4. Scale bar, 90 μm.
F  Quantification of vascular length and number of branchpoints; (n = 4). Results are presented as mean ± SEM and statistical significance was analyzed by Mann–Whitney test.

specific antibody (Fig 7F), consistent with data from the human protein atlas which detected COCO expression in the mitochondria (https://www.proteinatlas.org/) (Thul *et al*, 2017). These data suggest possible uptake and transport to mitochondria of exogenously added COCO.

**COCO alters the redox and glycolytic balance of endothelial cells**

To further explore the mechanisms underlying the effects of COCO on endothelial cells, we performed transcriptomic analysis of HUVECs stimulated with COCO for 16 h (Fig 8A). Among the most dysregulated genes (Log$_2$Fc ≥ ±1; *P*value ≤ 0.05) (Fig 8B), we found changes for several transcripts in regard to mitochondrial metabolic function and energy metabolism, such as Type 2 NADH dehydrogenase (*NDUFV2*), mitochondrial NAD kinase 1 (*NADKD1*), Ubiquinol-Cytochrome C Reductase Core Protein (*UQCRC1*), Oct-1 (*POU2F1*), and Transitional Endoplasmic Reticulum ATPase (*VCP*) among others (Fig 8B; Wang & Jin, 2010; Xu *et al*, 2017; Zhang *et al*, 2018; Parzych *et al*, 2019). Further gene set enrichment analysis (GSEA) also showed that COCO induced a significant down-regulation of pathways involved in TGFβ and BMP signaling in HUVECs, consistent with published data showing that

COCO acts as an antagonist of BMP and TGFβ signaling (Fig 8C). A significant increase in genes associated with generation of ROS was also found in COCO-treated HUVECs (Fig 8C). Given the changes that we observed in genes associated with mitochondrial metabolic function and ROS production, combined with the observation that TGFβ and BMP signature pathways, which have been involved in endothelial cell metabolism (Rodríguez-García *et al*, 2017; Lee *et al*, 2018), were decreased in COCO-treated cells, we addressed the effects of COCO on endothelial cell ROS generation and energy metabolism.

To better explore endothelial cell metabolism following COCO stimulation, we measured oxygen consumption rates (OCR) and extracellular acidification rate (ECAR), an indicator of glycolysis, in HUVECs stimulated with COCO for 1, 2, and 24 h. Treatments with 2-Deoxyglucose (2-DG), a potent inhibitor of glycolysis, severely decreased ECAR and, to a lesser extent oxygen consumption in HUVECs (Fig 9A). While we observed that COCO slightly decreased basal OCR at 2-h following stimulation, COCO-treated cells exhibited significantly reduced ECAR compared with control cells, suggesting decreased glycolytic capacity (Fig 9A). However, both OCR and ECAR levels were similar 24 h following COCO treatments, suggesting that these changes are transient. Endothelial cells rely primarily

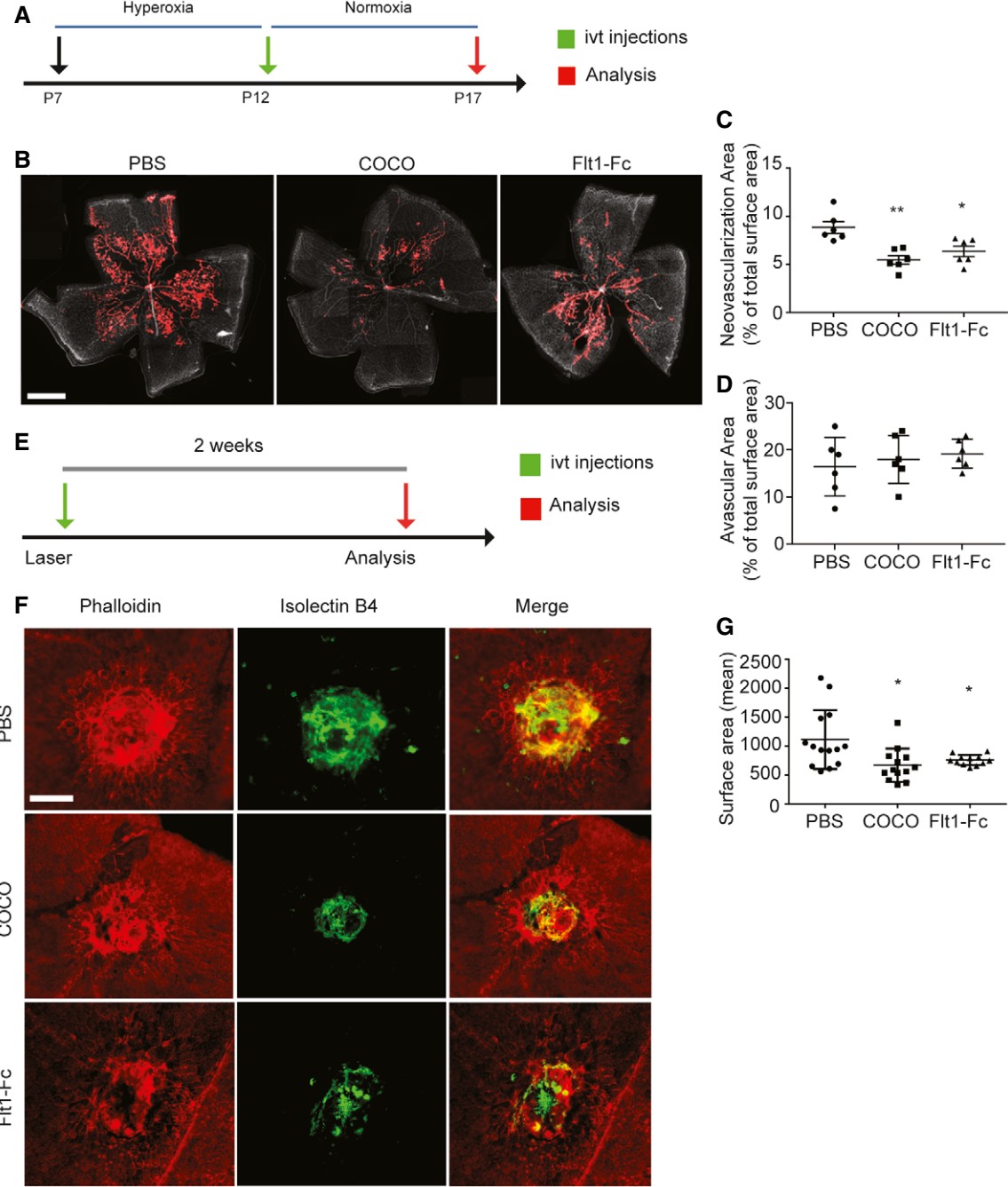

**Figure 5. COCO inhibits retinal and choroidal pathological angiogenesis.**

A   Schematic of the experimental strategy to assess the effects of COCO in the Oxygen-induced Retinopathy model.
B   Retinal flat mounts of P17 OIR mice injected with PBS, COCO, or Flt1Fc are stained with IB4. Red areas highlight vascular tufts. Scale bar, 500 μm.
C   Quantification of neovascular tuft area in P17 OIR. **$P = 0.0021$ (PBS vs. COCO); *$P = 0.0122$ (PBS vs. Flt1Fc); ($n = 6$).
D   Quantification of central avascular area in P17 OIR; ($n = 6$).
E   Schematic of the experimental strategy to assess the effects of COCO in the Laser-induced Choroidal Neovascularization (CNV) model.
F   Choroidal flat mounts of adult mice subjected two weeks prior to laser-CNV injected with PBS, COCO or Flt1Fc stained with IB4 (green) and phalloidin (red). Scale bar, 100 μm.
G   Quantification of CNV surface area. *$P = 0.0210$ (PBS vs. COCO); *$P = 0.0113$ (PBS vs. Flt1Fc); ($n = 12$–14 burns from 5 animals/group). Results are presented as mean ± SEM and statistical significance was analyzed by Mann–Whitney test. *$P < 0.05$, **$P < 0.01$.

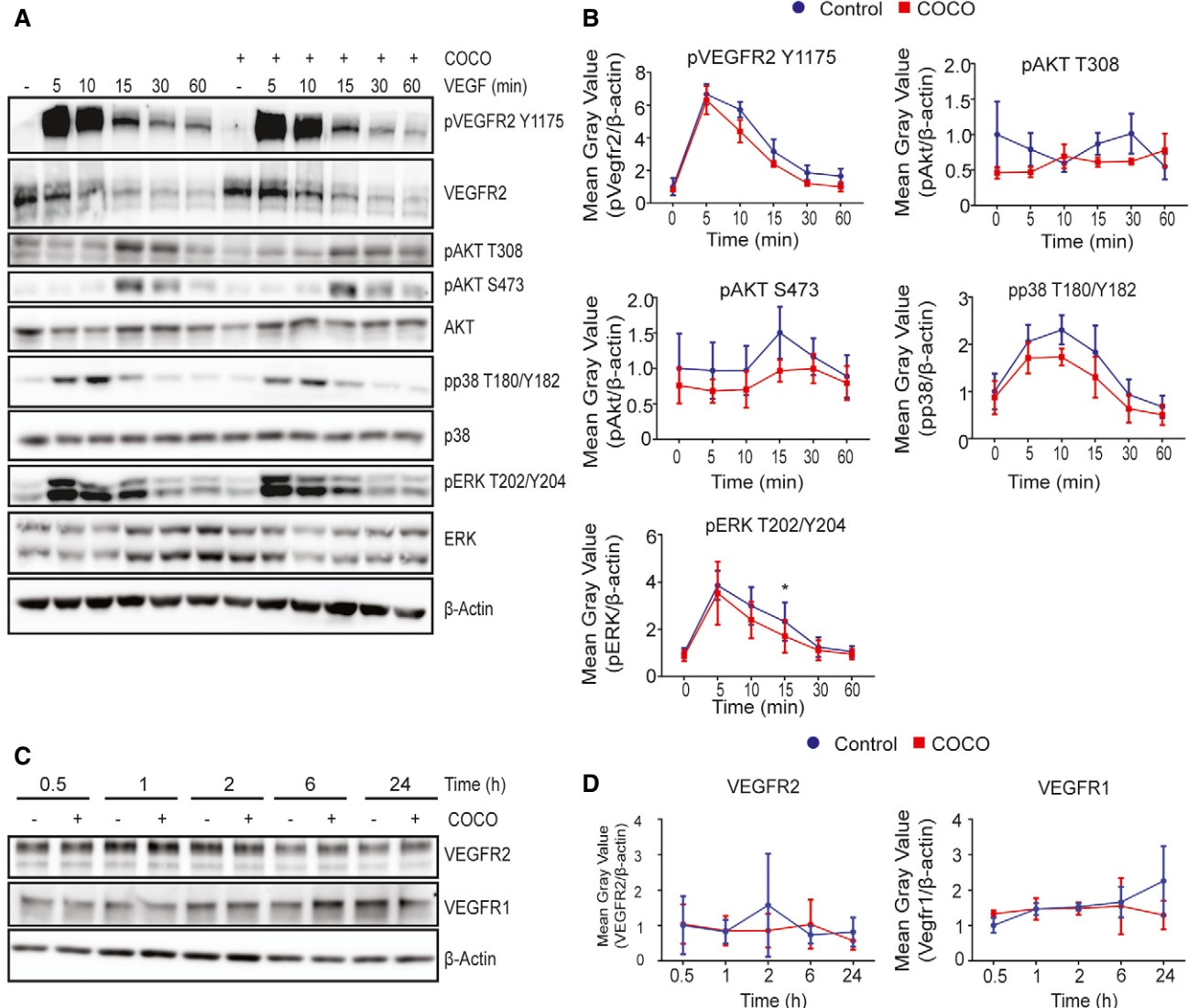

**Figure 6. COCO does not impair VEGF signaling in cultured endothelial cells.**

A   Western blot analysis of HUVECs treated 24 h prior with COCO and stimulated with VEGF for up to 60 min. Immunoblots are representative of 4 independent experiments.

B   Densitometric analysis of Western blot experiments. *$P$ = 0.0457; ($n$ = 4).

C   COCO treatment does not alter VEGF receptor expression in HUVECs. Immunoblots are representative of 3 independent experiments.

D   Densitometric analysis of Western blot experiments; ($n$ = 3). Results are presented as mean $\pm$ SEM and statistical significance was analyzed by two-way ANOVA. *$P < 0.05$.

Source data are available online for this figure.

on glycolysis as their main energy source and ATP production during angiogenesis (Draoui et al, 2017). Glucose uptake and metabolism are increased during angiogenesis to meet this energy demand. As glycolytic flux was reduced HUVECs, we also tested whether glucose uptake was altered in HUVECs treated with COCO. Analysis of glucose uptake in HUVECs revealed a rapid but transient decrease in glucose transport 1 h after COCO treatment (Fig 9B), which correlated with decreased ECAR. Total ATP levels were also assessed after 1, 6, and 24 h of treatment with COCO. ATP levels were rapidly reduced after COCO treatment and reached a minimum

of 25% of baseline levels after 1 h although these levels went up after 6 h (Fig 9C), indicating that COCO leads to a significant but transient inhibition of ATP production. Altogether, these data suggest that COCO transiently impairs glycolytic capacity and ATP production in endothelial cells, which may be reflected in the decreased proliferative and migrating capacity of endothelial cells treated with COCO.

The NAD+/NADH redox couple is known as a regulator of cellular energy metabolism such as glycolysis and mitochondrial oxidative phosphorylation (Xiao et al, 2018). As a co-factor for a

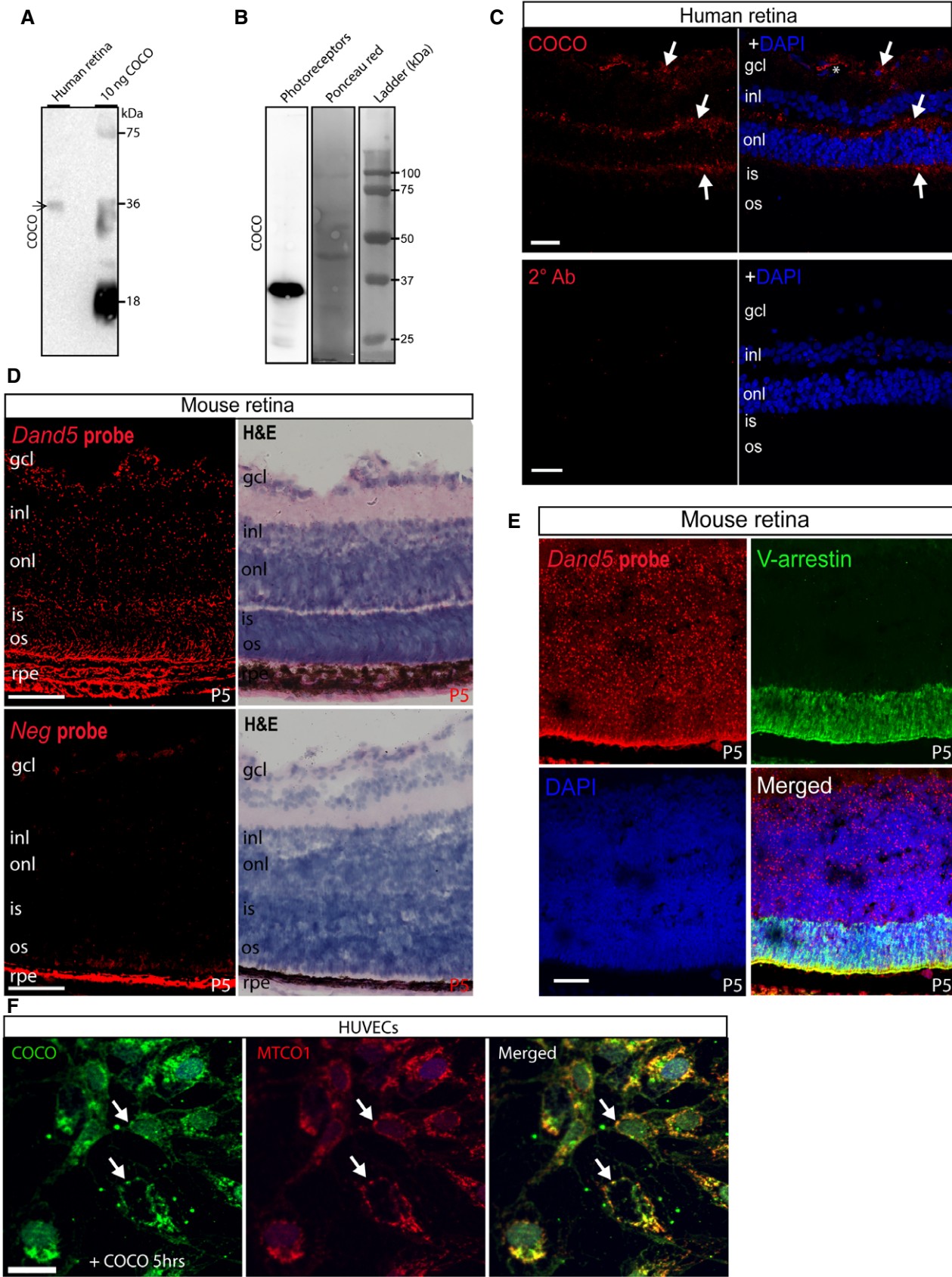

**Figure 7.**

Figure 7. COCO is expressed in human and mouse retina and localizes to mitochondria in COCO-exposed HUVECs.

A  Western blot of whole adult human retina extracts incubated with an anti-human COCO antibody and revealing a unique band at ~ 36 kDa (arrow). Recombinant human COCO was used as positive control.
B  Western blot of photoreceptors produced from human embryonic stem cells incubated with an anti-mouse COCO antibody and revealing a unique band at ~ 36–38 kDa (arrow). Western blots are representative of 3 independent experiments.
C  Immunofluoresence analysis of adult human retina sections with an anti-mouse COCO antibody. Specific immunoreactivity was observed in multiple areas (arrows) when compared with sections only exposed to the secondary antibody. Scale bar, 40 μm.
D  RNAscope in situ hybridization (Dand5, top; Negative probe (dapB); down) and hematoxylin staining of P5 mouse retinas. Scale bar, 40 μm.
E  Dual RNAscope in situ hybridization and visual-arrestin immunohistochemistry of P5 mouse retinas. Scale bar, 40 μm. Images are representative of 4 animals.
F  Immunofluoresence analysis of HUVECs exposed to COCO for 5 h prior to fixation. Exogenously added COCO was detected using an anti-human COCO antibody, showing co-localization with human mitochondria (MTCO1 antibody). Scale bar 25 μm.

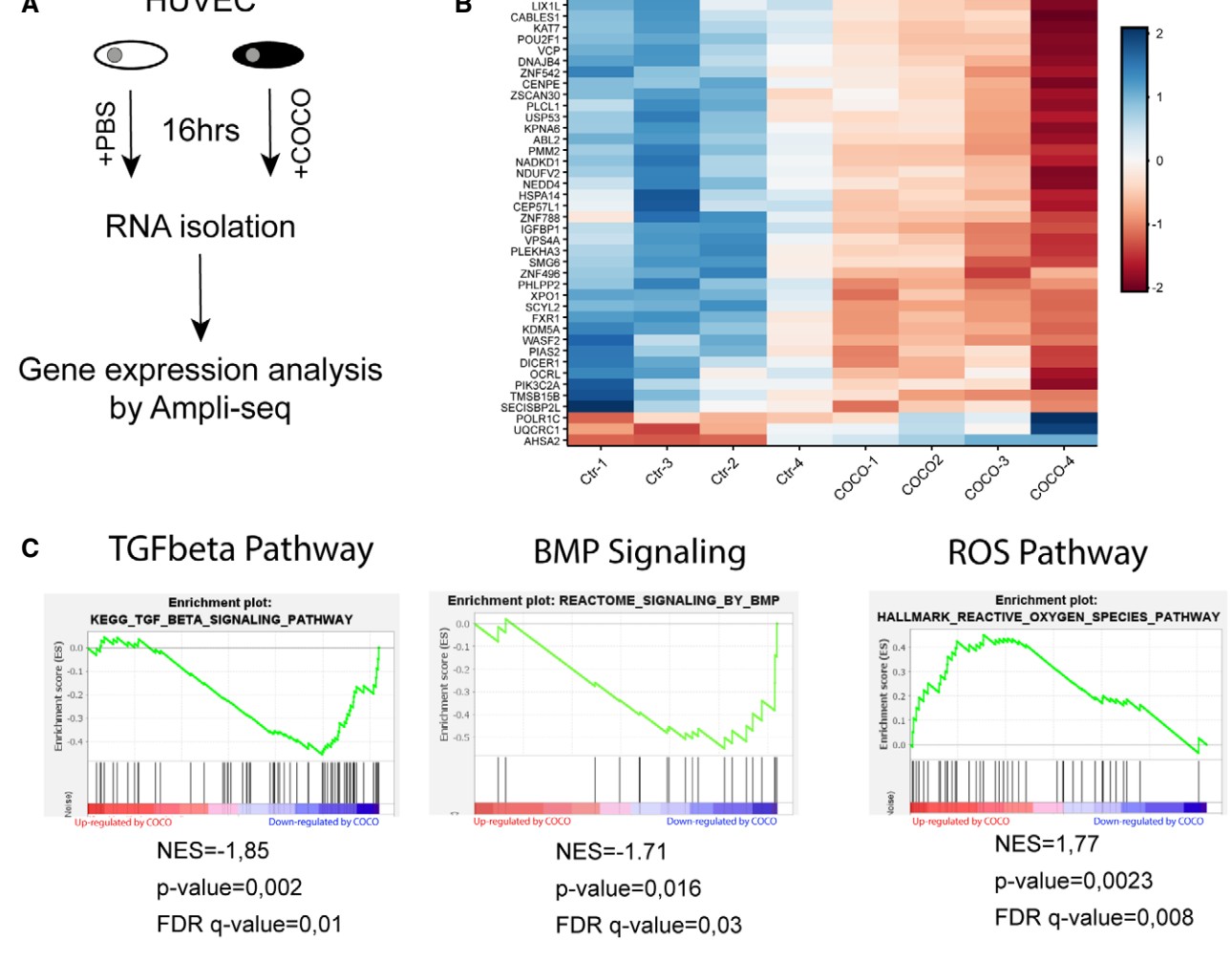

Figure 8. Transcriptional changes associated with COCO stimulation in ECs.

A  Scheme of the protocol to determine differential gene expression after treatment of HUVECs 16 h with COCO.
B  Differential gene expression heatmap generated by DESeq2 of top-altered genes of HUVECs treated with COCO compared with PBS-treated control cells (n = 4 independent samples/group).
C  GSEA pathway analysis of control vs. COCO-treated HUVECs for TGFβ, BMP, or ROS pathways.

variety of oxidoreductase enzymes, NAD(H) mainly functions in biodegradation reactions and energy generation. We therefore evaluated whether decreased ATP production in the presence of COCO was associated with alterations in the ratio of NAD+ and its reduced form. HUVECs treated with COCO displayed a significant decrease in the NAD+/NADH ratio (Fig 9D). NAD+, required for the ATP-generating steps of glycolysis, is regenerated from NADH by mitochondrial NADH dehydrogenase or lactate dehydrogenase

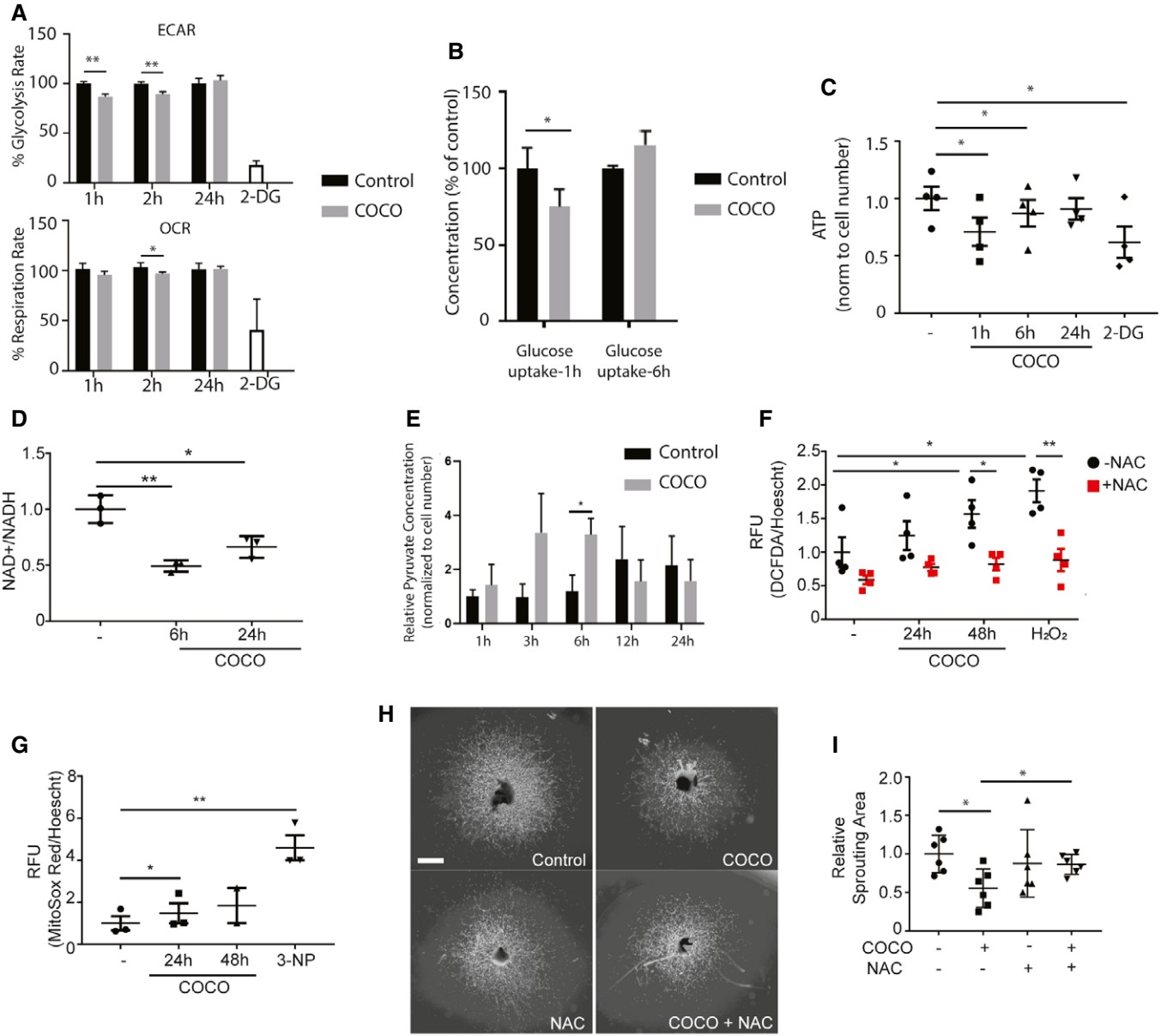

**Figure 9.  COCO alters the energy metabolism of endothelial cells.**

A   Seahorse analysis of mitochondrial respiration (OCR) and glycolysis (ECAR) in HUVECs stimulated with COCO for 1, 2, and 24 h. **$P = 0.0032$ for ECAR, 1 h; **$P = 0.0079$ for ECAR, 2 h; *$P = 0.0255$ for OCR, 2 h; ($n = 5$).
B   Media glucose levels (BioNova analysis) in HUVECs cultured with COCO for 1 or 6 h, expressed as % of initial levels *$P = 0.0459$; ($n = 4$).
C   Evaluation of ATP content in HUVECs stimulated with COCO for 1, 6, or 24 h. *$P = 0.0120$ (− vs. 1 h); *$P = 0.0109$ (− vs. 6 h); *$P = 0.0219$ (− vs. 2-DG); ($n = 4$).
D   Determination of NAD+/NADH ratio in HUVECs for 6 or 24 h with COCO. **$P = 0.0028$ (6 h); *$P = 0.0207$ (24 h); ($n = 3$).
E   Evaluation of pyruvate content in HUVECs stimulated with COCO for 1, 3, 6, 12, or 24 h. *$P = 0.0244$; ($n = 4$).
F   Determination of cellular ROS using DCFDA fluorescence in HUVECs treated with COCO and/or N-acetylcystein. *$P = 0.0343$ (− vs. 48 h-NAC); *$P = 0.0114$ (− vs. $H_2O_2$-NAC); *$P = 0.0252$ (48 h-NAC vs. 48 h + NAC); **$P = 0.0067$ ($H_2O_2$-NAC vs. $H_2O_2$ + NAC); ($n = 4$).
G   Determination of mitochondrial superoxide using MitoSox fluorescence in HUVECs treated with COCO. *$P = 0.0372$ (− vs. 24 h); *$P = 0.0137$ (− vs. 3-NP); ($n = 3$).
H   Representative images of choroidal explants cultured for 4 days in the presence or absence of COCO and/or N-acetylcystein. Scale bar, 500 µm.
I   Quantification of sprouting surface area of micrographs shown in (H). *$P = 0.0126$ (− vs. COCO-NAC); *$P = 0.0200$ (COCO-NAC vs. COCO + NAC); ($n = 6$). Results are presented as mean ± SEM and statistical significance was analyzed by Mann–Whitney test. *$P < 0.05$, **$P < 0.01$.

(Stambaugh & Post, 1966). Accordingly, we observed a significant increase in accumulation of intracellular pyruvate after 6 h of treatment with COCO (Fig 9E). Taken together, these data show that COCO treatments lead to an accumulation of pyruvate and decreased NAD+ regeneration, associated with a reduction in overall glucose consumption.

The build-up of NADH together with low ATP production has been associated with increased mitochondrial ROS generation. Indeed, the build-up of NADH due to reduced ATP synthesis and consequent lowered respiration rate decreases the NAD+/NADH ratio and leads to $O_2^{\bullet-}$ formation by complex I in the mitochondrial matrix (Kudin *et al*, 2004; Kussmaul & Hirst, 2006). Given that GSEA analysis revealed a significant increase in ROS-related genes (Fig 8C), levels of ROS were also evaluated in HUVECs treated for up to 48 h with 60 ng/ml COCO using the ROS probe 2′,7′–dichlo-rofluorescin diacetate (DCFDA), which generates fluorescence proportional to the amount of oxidized DCFDA to DCF. We observed that HUVECs cultured with COCO for 48 h showed increased levels of ROS, which were quenched by concomitant treatment with N-acetyl cysteine (NAC) (Fig 9F). Mitochondrial-specific superoxide production in HUVECs treated with COCO was also assessed using MitoSox Red dye, and we found that COCO significantly increased mitochondrial superoxide production after 24-h exposure (Fig 9G). Taken together, these data suggest that COCO stimulation may impair glucose-dependent energy production, which in turn may overload complex I with NADH leading to decreased ATP production and enhanced complex I production of ROS (Zorov *et al*, 2014).

The redox system is heavily involved in endothelial cell function and dysfunction. Indeed, while modest and controlled generation of ROS in endothelial cells is required for numerous vital signaling pathways involved in cell survival, proliferation, activation, stress response, cell motility, vasodilation, and angiogenesis, high levels of oxidants have inhibitory effects on endothelial cell function (Jerkic *et al*, 2012; Kim & Byzova, 2014) and have been associated with increased senescence and DNA damage. To evaluate whether antioxidants could modulate the effects of COCO on sprouting angiogenesis, choroidal explants were cultured with or without COCO, in the presence or absence of NAC. As previously described, control choroidal explants cultured in the presence of VEGF displayed numerous endothelial sprouts after 5 days of culture. Accordingly, the addition of COCO resulted in a significant inhibition of the choroidal sprouting area. The addition of NAC significantly abrogated the inhibitory effects of COCO on vascular sprouting, suggesting that increased ROS generation may also play a significant role in the anti-angiogenic effects of COCO (Fig 9H and I).

## Discussion

We have identified COCO, a secreted antagonist of BMP, TGFβ, and Wnt ligands, as a potent inhibitor of neovascularization in the eye. Injection of COCO during developmental retinal angiogenesis delayed formation of new blood vessels, and the effect of COCO on blood vessel formation appeared to be as potent as that of a VEGF inhibitor, Flt1Fc. In experimental models of choroidal neovascularization and vascular retinopathy, COCO also displayed potent inhibitory effects on angiogenesis. Importantly, COCO was shown to act specifically on developing blood vessels, as intravitreal delivery in adult mice did not result in obvious morphological changes on the mature vasculature. Also, long-term delivery of COCO, while having a potent effect on newly formed blood vessels, did not adversely affect photoreceptors. Few studies have looked at the effects of Dan family members during vascular development. Dan family members

have been shown to act as antagonists of BMP, TGFβ, and Wnt signaling molecules. As these pathways play critical roles in vascular development, it is not unexpected that COCO, by antagonizing their signaling, may interfere with the angiogenic process. Specifically, COCO has previously been shown to inhibit BMP4 and Wnt1 in a dose-dependent manner in cultured photoreceptors (Zhou *et al*, 2015). In microvascular endothelial cells, BMP4, along with BMP2, have been shown to induce tube formation as well as promote migration (Rothhammer *et al*, 2007). Wnt1 has also been shown to induce the proliferation and survival of human endothelial cells, is expressed in developing blood vessels, and exerts salutary effects on postnatal endothelial progenitors (Masckauchan *et al*, 2005; Gherghe *et al*, 2011). By interfering with Wnt, BMP, and TGFβ signaling, COCO may therefore interfere with blood vessel development by altering several components of the angiogenic response, including proliferation and migration of endothelial cells.

One of the more surprising findings of this study was the observation that COCO affects energy metabolism in endothelial cells. Several studies have highlighted the importance of metabolic regulation in the endothelium and shown the critical importance of metabolic and glycolytic pathways in driving the angiogenic process (De Bock *et al*, 2013; Treps *et al*, 2016; Cantelmo *et al*, 2016; Eelen *et al*, 2018). Experimental data show that glycolytic ATP is a driver for endothelial cell rearrangements in the sprout by enhancing filopodia formation (Eelen *et al*, 2018) and that inhibition of the glycolytic flux blocks both physiological and pathological angiogenesis (Treps *et al*, 2016). Our data therefore argues that COCO, which reduces endothelial cell metabolic activity, may lead to a quiescent phalanx cell-like phenotype (Potente *et al*, 2011). The mechanisms by which COCO alters the energy metabolism and redox homeostasis of endothelial cells remain to be established. Our transcriptomics data reveal that several genes associated with mitochondrial function are altered in COCO-treated HUVECs, such as *NDUFV2*, a subunit of mitochondrial complex I, and *UQCRC1*, a component of the mitochondrial complex III, which could play a role in the effects of COCO on cellular metabolism and ROS production. Indeed, decreased expression or activity of *NDUFV2* has been associated with impaired complex I activity, correlated with reductions in ATP synthesis and mitochondrial membrane potential, while disruption of *UQCRC1* results in decreased complex III formation and activity, mitochondrial membrane potential, and ROS formation (Ogura *et al*, 2012; Shan *et al*, 2019; Wang *et al*, 2019; Chen *et al*, 2020). Therefore, COCO may alter endothelial metabolism and ROS production in part by modulating the expression of genes involved in mitochondrial function, although the rapid onset of the effects of COCO on glucose uptake and ATP production (starting 1 h after stimulation) is also suggestive of mechanisms that are independent from the modulation of gene expression. Indeed, the observation that exogenously added COCO rapidly triggers metabolic changes and localizes to the mitochondria in endothelial cells seems to suggest that it may also interact with unidentified cell surface or intracellular mediators involved in mitochondrial function. The interaction of Dan family members with cell surface receptors is indeed not unprecedented. Gremlin, a BMP antagonist structurally related to COCO, has previously been shown to interact with VEGFR2, although its function as an agonist or antagonist of VEGF signaling is still being debated (Mitola *et al*, 2010; Grillo *et al*, 2016; Dutton *et al*, 2019). Furthermore, Gremlin has also been shown to interact with Slit2 and block

Robo signaling (Tumelty et al, 2018), which is an important mediator of vascular development in the retina (Rama et al, 2015). Gremlin-BMP2 complex have also been shown to have a high affinity to heparan sulfates, whose presence on the cell surface and in the extracellular environment is critical to many physiological processes including angiogenesis (Kattamuri et al, 2017). Further studies investigating the interactions of COCO with alternative binding partners and the implications of its intracellular transport and its presence in the mitochondria will bring a better understanding of its role in cell metabolism and vascular development. Taken together, our data indicate that COCO may alter the endothelial glycolytic flux, whereby redox balance between NADH and NAD+ is perturbed and may result in decreased ATP production and increased ROS generation. This balance in energy production likely plays a significant role in the anti-angiogenic effects of COCO.

The effects of COCO on endothelial cell metabolism could also be linked to its ability to sequester Wnt, TGFβ, and/or BMP ligands. Indeed, studies have previously demonstrated that Wnt signaling is associated with altered glycolytic metabolism through PDK1-mediated inhibition of pyruvate flux to mitochondrial respiration (Pate et al, 2014). Wnt-driven Warburg metabolism characterized by increased reliance on glycolysis for energy production has also been reported in several cell lines (Esen et al, 2013; Pate et al, 2014). Furthermore, BMP signaling can also regulate glucose metabolism in chondrocytes through the upregulation of the glucose transporter Glut1 (Lee et al, 2018). Pulmonary hypertension, defined by altered BMP signaling, is also associated with endothelial dysfunction, an imbalance of proliferation and apoptosis, and an altered glycolytic metabolic profile (Tuder et al, 2013; Goumans et al, 2018). Similarly, depending on the context, TGFβ stimulation can also modulate mitochondrial respiration, glycolysis, and ROS generation (Abe et al, 2013; Bernard et al, 2015; Soukupova et al, 2017). As such, COCO may act in part by interfering with Wnt, TGFβ, and/or BMP signaling in endothelial cells, leading to alterations in energy production. Given that we report a significant inhibition of TGFβ pathway, along with a reduction in the expression of several genes involved in BMP signaling in COCO-treated HUVECs (Fig 8), we anticipate that COCO may modulate the energy metabolism of endothelial cells at least in part through the sequestration of TGFβ and/or BMP ligands. Notably, COCO appeared to act independently of BMP9-BMP10/Alk1 signaling, as shown by the observation that COCO retained its anti-angiogenic properties in Alk1ΔEC mice (Fig EV5), which lack Alk1 specifically in the endothelium. This does not preclude however that COCO could act by inhibiting the signaling of other BMP ligands, such as BMP2 and BMP4 (Mausner-Fainberg et al, 2019).

We show here a novel effect of COCO delivery in retinal vascular development. While we demonstrate the anti-angiogenic effects of exogenous COCO, the contributions of endogenous COCO in retinal angiogenesis remain to be demonstrated. Its inhibitory effects on neo-vessels, combined with its expression in the neural retina, may suggest a role for the prevention of angiogenesis in the outer retina and the maintenance of the photoreceptor avascular privilege. Our study has also important implications for the development of therapies targeting neovascular diseases in the eye, as we describe a novel inhibitor of pathological retinal and choroidal angiogenesis that apparently acts independently of VEGF signaling. Importantly, work is under progress to test whether chronic intra-ocular injection

of COCO at therapeutic concentrations can lead to local and peripheral toxic effects in animals. While the mechanisms underlying the effects of COCO on blood vessels remain to be fully elucidated, our data show that its modulation of metabolism may play an important role in its effects on endothelial cells. Future studies will help decipher the mechanisms by which COCO affects the formation of new blood vessels.

# Materials and Methods

### Mice, tissue samples, and reagents

This study was conducted according to the CIUSS-de-l'Est-de-l'Ile-de-Montreal institutional guidelines and the Declaration of Helsinki. Eyes from human donors were provided by our local Eye Bank (Banque de tissus oculaires pour la recherche en vision; Centre de recherche du CHU de Québec-Université Laval; Quebec City, Quebec, Canada). Adult (3-month-old) and P1 to P17 pups mice C57BL/6J (Jax Mice) or Alk1ΔEC mice (Aspalter et al, 2015) (kindly provided by Paul S Oh (University of Florida)) were used in this study. All animals were housed and bred in a normal experimental room and exposed to a 12 h light/dark cycle with free access to food and water. All animal procedures were conducted under the regulation of Canadian federal and institutional guidelines (Protocol #2020-1921). Human Umbilical Vein Endothelial Cells (HUVECs; Promocell) were cultured in EndoGro-VEGF medium (EMD Millipore). Human Retinal Microvascular Endothelial Cells (HRMECs; Cell Systems, Kirkland, USA) were grown in EGM-2 microvascular medium (Lonza). Cells were routinely tested for mycoplasma contamination. COCO (R&D systems; Cat#3047-CC) was resuspended according to the instructions of the manufacturer at a stock concentration of 100 µg/ml.

### Intravitreal Injections

Animals were anaesthetized with isofluorane. A 10 µl Hamilton syringe with a glass-pulled capillary was inserted with a 45° injection angle into the vitreous. When accessing the role of COCO developmentally, animals were injected either at P1 (final concentration COCO: 100 ng/ml in 5 µl vitreous; or similar volume PBS) or P3 to evaluate vascular growth (Schmucker & Schaeffel, 2004). During the neovascularization phase of OIR, animals were injected at P12 (2 µl) before sacrifice at P17 for quantification of neovascularization. For adult mice and laser-induced neovascularization, intravitreal injections were performed under a surgical microscope. Mice were anaesthetized with isofluorane. Pupils were dilated using 1% tropicamide and a 33-gauge needle was inserted from the limbus with a 45° injection angle into the vitreous.

### Choroidal sprouting assays

Choroidal sprouting assays were performed as previously described (Shao et al, 2013). Following the removal of neuroretina from the posterior pole, the complex consisting of the retinal pigment epithelium (RPE)-choroid-sclera was collected, cut in 16 explant fragments, and cultured in Matrigel (BD biosciences) in 24-well plate. Explants were stimulated at day 1 and day 3 with COCO (100 ng/ml) and

sprouts were imaged at day 5. Quantification of sprouting area was performed using software analysis (Fiji/ImageJ).

## Oxygen-induced retinopathy

C57BL/6J mouse pups at postnatal day (P)7 and their fostering mothers (CD1, Charles River) were subjected to 75% oxygen in an oxycycler chamber for 5 days. Pups were then returned to normoxia at P12 and administered 100 ng/ml (final concentration) of recombinant human COCO intravitreally or similar volume of vehicle in the contralateral eye. Eyes were enucleated at P17 and processed for immunostaining.

## Laser-induced choroid neovascularization

Three-month-old C57BL/6J mice were anesthetized with a ketamine/xylazine mix prior to applying a photocoagulating laser (400 mW intensity, 0.05 s exposure time). Four spots were burned around the optical nerve. Mice received 100 ng/ml (final concentration) of recombinant human COCO intravitreally or similar volume of vehicle in the contralateral eye. Eyes were enucleated after 14 days and processed for immunostaining.

## Immunohistochemistry

Ocular globes were initially fixed for 15 min in 4% paraformaldehyde (PFA). Retinas or choroids were collected after eyes dissection in PBS and blocked 1 h in PBS 3% BSA 0.1% Triton X-100. Fixation was prolonged in 1% PFA overnight for choroid extraction or eyes sectioning. Prior to sectioning, eyes were maintained in sucrose gradients (10–30%), cryo-preserved in a matrix gel, and sliced in 14 μm sections on a cryostat (Leica CM3050S). Staining with either FITC-labeled isolectin GS IB4 (Life technologies corporation), rhodamine phalloidin (Cedarlane Laboratories), phospho-histone H3 (Abcam), Collagen IV (Abcam), or cleaved caspase-3 (Cell Signaling) antibodies were performed on whole and/or sectioned retinas/choroids (see Table EV1 for the list of antibodies). Retinas and choroids were then mounted in fluoromount aqueous medium (Sigma-Aldrich). Quantitative analysis of tufts, vaso-obliterated, or vessel areas were performed using ImageJ/Swift_NV as previously described (Ntumba *et al*, 2016). Neovascular tuft formation was quantified by comparing the number of pixels in the affected areas with the total number of pixels in the retina. The avascular area in the retina was measured in the same way.

For HUVEC immunostaining, cells stimulated for 5 h with recombinant COCO were subsequently acid-washed to strip the surface-bound molecules. Cells were fixed for 15 min in 4% PFA and blocked 1 h in PBS 3% BSA 0.1% Triton X-100, followed by overnight incubation with primary antibodies (anti-hCOCO, Sigma; anti-MTCO1, Abcam). Cells were then washed, incubated with secondary reagents for 1 h at room temperature, washed and mounted. Slides were then mounted with fluoromount containing DAPI (Sigma) and visualized by confocal microscopy (Olympus Fluoview).

## Endothelial cell sprouting assays

Sprouting assays were performed as previously described (Larrivee *et al*, 2012). Briefly, HUVECs or HRMECs (250,000 cells/well in 6-well plates) were resuspended in 300 μl fibrinogen solution (2.5 mg/ml fibrinogen, Sigma-Aldrich) in EBM-2 (Lonza) supplemented with 2% FBS and 50 μg/ml aprotinin (Sigma-Aldrich), and plated on top of a pre-coated fibrin layer (400 μl fibrinogen solution clotted with 1 U thrombin (Sigma-Aldrich) for 20 min at 37°C). The second layer of fibrin was clotted for 1 h at 37°C. NHDF cells (250,000 cells/well), in EBM-2 supplemented with 2% FBS and 25 ng/ml VEGF, with or without COCO, were then plated on top of the fibrin layers. Cultures were incubated at 37°C, 5% $CO_2$. Quantification of sprouting area was performed using software analysis (Fiji/ImageJ).

## Transcriptome analysis by AmpliSeq

Transcriptomic analysis of HUVEC cells treated with or without COCO for 16 h, in quadruplicate, was performed using the Ion AmpliSeq Transcriptome Human Gene Expression Kit (Thermo Fisher Scientific) according to manufacturer's instructions. Briefly, mRNA from 10 ng of total RNA was reverse transcribed using SuperScript VILO cDNA Synthesis Kit (Thermo Fisher Scientific) and amplified with Ion AmpliSeq HIFI Mix together with primers from the Ion AmpliSeq Transcriptome Human Gene Expression Core Panel simultaneously targeting over 20,000 RefSeq genes. Primer sequences were partially digested with FuPa Reagent and then barcoded using Ion Xpress Barcodes (Thermo Fisher Scientific). Purification was carried out by AMPure XP Reagent (Beckman Coulter). Libraries concentrations were defined by qPCR using Ion Library Quantification kit (Thermo Fisher Scientific). Libraries were pooled together for emulsion PCR, carried out using the Ion Chef Instrument (Thermo Fisher Scientific). Purified Ion Sphere Particles were loaded on Ion P1 Chip. The sequencing was performed on Ion Proton system (Thermo Fisher Scientific). Ion Torrent software, Torrent Suite v5.12 (Thermo Fisher Scientific), was used for base calling, alignment to the human reference genome (hg19) and quality control. Raw reads were then analyzed automatically using the AmpliSeqRNA plugin to generate gene-level expression values. Differential gene expression was determined using DEseq2 (version 3.11) package. Gene set enrichment analysis (GSEA) was performed using the GSEA software using pre-defined gene sets based on prior biological knowledge (version 4.1).

## Scratch assays

Confluent HUVEC monolayers were grown in 6-well plates. Cells were starved 18 h in EBM-2 medium with 1% FBS. A horizontal wound was created using a sterile 200 μl pipette tip. Next, the cells were washed with EBM2 at 37°C and incubated in EBM-2 supplemented with VEGF-A (25 ng/ml) with or without COCO (60 ng/ml) at 37°C for 16 h. Pictures of scratch wounds were taken just before stimulation (time 0) and after 16 h. Migration % was calculated using ImageJ software.

## Flow cytometry

Sub-confluent HUVECs were cultured overnight in starvation medium (EBM2-1%FBS) in the presence or absence of 60 ng/ml COCO, followed by VEGF stimulation for 1 h. HUVECs were subjected with a pulse of 5-ethynyl-2′-deoxyuridine (EdU) for 1 h,

and flow cytometry analysis of EdU incorporation was performed as previously described (Oubaha *et al*, 2016). Detection of apoptotic cells was performed using a dead cell apoptosis kit (Cell Signaling) according to the instructions of the manufacturer.

## Western blotting

Cells were washed with cold PBS and extracted in Laemmli's buffer, followed by sonication. Samples were run on SDS–PAGE gels and transferred onto nitrocellulose membranes. Membranes were blocked with 5% Bovine Serum Albumin (BSA) and probed with primary antibodies overnight at 4°C. HRP-conjugated secondary antibodies (Vector Laboratories) were used to detect primary antibodies. Antibodies are listed in Table EV1. Densitometric band intensity quantification of detected immunoblotting protein-antibodies complexes was done using ImageJ software.

## RNAscope *in situ* hybridization

*In situ* hybridization was performed on paraformaldehyde-fixed, OCT-embedded sections as directed by the manufacturer (Advanced Cell Diagnostics, Hayward, CA, USA). We used RNAscope Probe-Mm-Dand5 (NPR-0006197) to detect mouse *Dand5*. The *Dand5* probe was designed to target 53–1,175 of mouse *Dand5* (NR_033145.1). Negative control sections were probed for bacterial dihydrodipicolinate reductase mRNA (dapB).

## ROS measurement

HUVECs were plated into a black 96-well plate and treated with COCO, $H_2O_2$, or 3-NP. Media was removed after specific times and replaced with complete media containing 20 μM 2,7-Dichlorofluorescin diacetate (Sigma, D6883) for 30 min or 5 μM MitoSox Red (Life Technologies M36008) for 5 min at 37°C. Media was removed and cells were washed once with PBS. Fluorescence was measured at Excitation/Emission 485/535 nm (DCFDA) or 544/590 nm (MitoSox Red). After the initial reading, cells were incubated with Hoescht and fluorescence was detected at 355/460 nm. DCFDA and MitoSox fluorescence was normalized to Hoescht for each well.

## NAD+/NADH enzymatic cycling assay

Evaluation of cellular NAD+/NADH levels were performed as previously described (Kato *et al*, 1973; Lin *et al*, 2001). HUVECs plated into 100 mm dishes were treated with COCO for various time points. NAD+ and NADH were extracted using ice cold alkali (0.5 M NaOH, 1 mM EDTA) and acidic buffers (0.1 M HCl), respectively. Extracts were heated at 60°C for 30 min and buffers were neutralized with either the NADH (100 mM Tris–HCl pH 8.1, 0.05 M HCl) or NAD+ (0.4 M Tris) neutralization buffers. To attain measurable quantities of NAD+ or NADH, an amplifying cycling assay was performed. Extracts and NAD+ standards were incubated with cycling reagent (67 mM Tris–HCl pH 8, 200 mM EtOH, 1.3 mM beta-mercaptoethanol, 0.01% BSA, 2 mM oxaloacetic acid, 0.5 μg/ml malate dehydrogenase, 5 μg/ml alcohol dehydrogenase) for 1 h at room temperature. All samples were heated for 5 min at 100°C to stop enzymatic reactions then cooled on ice. For detection, extracts were incubated in an indicator buffer (50 mM 2-amino-2-methyl-

### The paper explained

**Problem**

Neovascular age-related macular degeneration (AMD) is a significant cause of vision loss in aging populations. Current therapies for neovascular AMD focus on the inhibition of a protein, Vascular Endothelial Growth Factor (VEGF), to block the growth of blood vessels in the eye. While therapies targeting VEGF have been shown to slow or stop the progression of AMD, there are still patients that show limited response to anti-VEGF drugs, and some adverse effects have also been reported after long-term treatments. Therefore, there is a need to identify alternative targets to block neovascularization in the eye in order to develop new therapeutic approaches.

**Results**

We have identified a protein, COCO, that can block the growth of blood vessels in the eye. Using murine models of developmental and pathological neovascularization, we have shown that COCO delays the growth of new blood vessels in the retina and the choroid, without affecting pre-existing mature vessels.

Mechanistically, using endothelial cells and cultures of choroidal explants, we have been able to demonstrate that COCO prevents the migration and proliferation of endothelial cells, in part by decreasing the signaling of TGFβ and BMPs, a family of genes known to regulate vascular growth, and by limiting energy metabolism and promoting reactive oxygen species production.

**Impact**

Our study identifies a new function for COCO, a factor that could be used in the clinics to limit the growth of pathological blood vessels in the eye, and could lead to the development of new therapies for the treatment of neovascular eye diseases.

propanol pH 9.9, 200 μM NAD+, 10 mM glutamate, 0.04% BSA, 5 μg/ml malate dehydrogenase, 2 μg/ml glutamate oxaloacetate transaminase) for 10 min at room temperature. Then, 100 μl if sample was transferred to a black 96-well plate. Fluorescence was detected using a plate reader (TECAN) at an excitation of 365 nm and emission of 460 nm. Standard curves were generated and the concentration of NAD+ and NADH were calculated from the standard curve.

## ATP detection

HUVECs were plated into 100 mm dishes and treated with COCO for various time points. ATP was measured by luminescence (ATP Detection Assay Kit; Cayman Chemicals).

## Seahorse analysis

The Seahorse analyzer XF96 (Agilent, Santa Clara, USA) was used to continuously monitor OCR and ECAR. Two days prior to the experiment, 2,000 cells/ well were seeded in a XF96 cell culture plate in EndoGro medium (EMD Millipore) and cultivated at 37°C in humidified atmosphere with 5% $CO_2$. Cells were then treated with COCO for 1, 2, and 24 h. One day prior to the experiment, 200 μl of XF calibrant was added to each well of the utility plate and the XF cartridge was incubated overnight at 37°C in a humidified atmosphere in a non-$CO_2$ incubator. Just before readings were performed, cells were washed twice with unbuffered XF DMEM

assay medium (containing 5 mM glucose and 2 mM glutamine, pH 7.4) and incubated for 1 h in a humidified non-$CO_2$ incubator. Inhibitor compounds were prepared as per the instructions of the Seahorse XF Cell Energy Phenotype Test Kit (#103325-100). Oligomycin and FCCP were added together to the injection port of the XF cartridge to yield final concentrations of 1 and 1.5 μM, respectively. After 15 min equilibration time, baseline OCR and ECAR were assessed every minute (after 3 min mixing, 2 min wait, 3:30 min measure). Five measurements were taken under stressed conditions following the compound injection. Following ECAR and OCR measurements, cells were lysed and protein concentration was determined using Pierce BCA protein assay kit (Thermo Fisher Scientific) for normalization. Data are presented as percentage of untreated cells.

### Metabolite measurements

HUVECs were cultured for up to 48 h in the presence or absence of COCO. Culture media was collected at 1, 3, 6, 12, 24, and 48 h, centrifuged at 16,260 *g* for 5 min, aliquoted, and stored at −80°C. Media was analyzed for glucose concentration using a BioProfiler 400 Analyzer (BioNova). For measurements of pyruvate, the culture media was analyzed using the Pyruvate Colorimetric/Fluorometric Assay Kit (BioVision) as per the manufacturer's instructions.

### Statistical analyses

Data analyses were performed in a blinded fashion. In animal studies, sample size was determined based on previous experiments performed in our laboratory, accounting for potential side effects of injections (weight loss, ocular inflammation). Animals were excluded from studies only if they displayed severe weight loss (over 20% body weight), or if they showed signs of ophthalmic inflammation following intravitreal injections. No randomization was used; however, we ensured that mice of similar weight and size were used for each group before treatments. All data are shown as mean ± standard error of the mean (SEM). Statistical analyses were performed for all quantitative data using Prism 6.0 (GraphPad). Statistical significance for paired samples and for multiple comparisons was determined by Mann–Whitney test and ANOVA, respectively. Normality and homogeneity were proved with Shapiro–Wilk test. Data were considered statistically significant if the *P* value was less than 0.05.

## Data availability

The datasets produced in this study are available in the following databases: Microarray: Gene Expression Omnibus GSE160099 (http://www.ncbi.nlm.nih.gov/geo/query/acc.cgi?acc = GSE160099).

**Expanded View** for this article is available online.

## Acknowledgements

The work described in this paper was supported by the Canadian Institutes of Health Research (FRN 363540; G.B, B.L), the Foundation Fighting Blindness (G.B, B.L), the Fonds de Recherche en Ophtalmologie de l'Université de Montréal and Hopital Maisonneuve-Rosemont Foundation. N.P was supported by a Suzanne Véronneau-Troutman scholarship and Fonds de Recherche en Ophtalmologie de l'Université de Montréal. B.L. was supported by a New Investigator Award from the Heart and Stroke Foundation of Canada. The authors wish to thank Houda Tahiri and Mikhail Sergeev for their help with the choroidal explant model and microscopy, respectively. The authors also thank Sergio Garcia-Crespo for his critical assessment of the manuscript. We kindly thank Ralf Adams and S. Paul Oh for Alk1ΔEC mice.

## Author contributions

Manuscript writing: BL, NP, EH, and GB; Guidance in designing of hypothesis and experiments: BL and GB; *In vivo* experiments, flow cytometry, choroidal sprouting assay, RNAscope, and immunostainings: NP; Transcriptomics analysis: NP, AF, and MB; *In vitro* metabolic experiments and VEGF signaling experiments: EH; Immunofluorescence on human retina and photoreceptors: AB; OIR and laser-CNV experiments: FP; Project conception: BL and GB.

## Conflict of interest

G.B. and A.F. are co-founders and shareholders of StemAxon™. G.B. and B.L. are inventors on patent application (U.S. Provisional Application No. 62/879,755) that covers the use of COCO for the treatment of ocular neovascularization. The rest of authors declare no conflict of interest.

## For more information

i   https://www.proteinatlas.org/
ii  https://crhmr.ciusss-estmtl.gouv.qc.ca/en
iii http://visionnetwork.ca/

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
