## [Review Process File · EMBO Molecular Medicine]

COCO/DAND5 inhibits developmental and pathological ocular angiogenesis

Natalija Popovic, Erika Hooker, Andrea Barabino, Anthony Flamier, Frédéric Provost, Manuel Buscarlet, Gilbert Bernier, and Bruno Larrivee

DOI: [10.15252/emmm.202012005](https://doi.org/10.15252/emmm.202012005)

Corresponding authors: Bruno Larrivee (blarrivee.hmr@ssss.gouv.qc.ca) , Gilbert Bernier (bernierg2@gmail.com)

Review Timeline:

Submission Date:	21st Jan 20
Editorial Decision:	14th Feb 20
Revision Received:	30th Oct 20
Editorial Decision:	18th Nov 20
Revision Received:	22nd Dec 20
Accepted:	24th Dec 20

Editor: Zeljko Durdevic

Transaction Report:

14th Feb 2020

Dear Dr. Larrivee,

Thank you for the submission of your manuscript to EMBO Molecular Medicine. We have now heard back from the referees whom we asked to evaluate your manuscript. As you will see from the reports below, the referees acknowledge the interest of the study. However, they raise some concerns that should be addressed in a major revision of the present manuscript. In particular, the characterization of the endogenous expression and function of COCO in the retina, as well as in-depth analysis of the transcriptomic data should be performed. Furthermore, assessing whether COCO exhibits additive effect with the inhibition of the VEGF signaling in vivo would increase the therapeutic relevance of COCO as an angiogenesis inhibitor.

We would therefore welcome the submission of a revised version within three months for further consideration and would like to encourage you to address all the criticisms raised to improve conclusiveness and clarity. Acceptance of the manuscript will entail a second round of review. Please note that EMBO Molecular Medicine encourages a single round of revision only and therefore, acceptance or rejection of the manuscript will depend on the completeness of your responses included in the next, final version of the manuscript. For this reason, and to save you from any frustrations in the end, I would strongly advise against returning an incomplete revision.

***** Reviewer's comments *****

Referee #1 (Remarks for Author):

The paper by Popovic and colleagues examines the involvement of COCO/DAND5 in developmental and pathological angiogenesis in the eye. Using HUVECs and a choroidal explant model, the authors show that COCO inhibits sprouting, migration and proliferation of cultured endothelial cells by altering their metabolic and redox status. These findings are further supported by a range of in vivo studies demonstrating that intravitreal injection of COCO inhibits retinal vascularisation during developmental and pathological angiogenesis in mouse models of retinopathy of prematurity and wet AMD. The paper is well written and interesting, but the conclusions could be strengthened in some areas.

1. The observation that COCO modulates ATP production, glucose uptake and redox signalling in HUVECs is interesting. However, it remains unclear if these changes result from inhibition of Wnt, BMP or TGFbeta signalling or are a direct consequence of its uptake into the cells. Addressing this issue would help to better define mechanistically how COCO modulates the angiogenic process and greatly improve the overall conclusions of the manuscript.
2. The in vitro data shows that COCO inhibits angiogenesis through a mechanism that is independent of VEGF signalling. Therefore COCO should have an additive effect with Flt1Fc in the in vivo models of retinal and choroidal angiogenesis. From a therapeutic perspective, this would seem important to confirm using at least one of the in vivo models.
3. It remains unclear from the paper whether COCO plays any endogenous role in modulating developmental or pathological angiogenesis in the eye. All of the experiments involve intravitreally injecting the recombinant protein.
4. There appears to be a major discrepancy between the concentrations of COCO used in the in vitro and in vivo experiments. For the in vitro experiments, concentrations up to 100ng/ml were used. For the mouse studies, it seems that 100ng was intravitreally injected into the mouse eye, which has a vitreal volume of around 5ul. It would be helpful if the authors could clarify the final vitreal concentration of COCO used, and if they have observed effects in vivo at more physiologically relevant protein concentrations.

Minor Comments:

1. P15 - How was the normality of the data tested prior to applying parametric statistics?
2. Fig 1E - The wound closure image shown for COCO appears to suggest that it is stimulating cell migration rather than inhibiting it.
3. How was the apoptosis data in Fig 1I,J statistically analysed?
4. In Fig 2B it appears that COCO is causing considerable inflammatory cell infiltration into the retina. It would be helpful if the authors could comment on this.
5. In Figs 2C and 5F, were the effects of COCO significantly greater than those of FltFc as suggested in the main manuscript text? If not, the text should be adjusted accordingly.
6. Data in Figs 2D and E would benefit from quantification to enable firm conclusions to be drawn.
7. In Fig 2F its unclear how changes in photoreceptor morphology could be determined, as suggested in the main text. Further clarification is needed here.
8. Fig 5B - in addition to reporting the neovascular area in OIR experiments, it would be normal practice to also present data on avascular areas. This would provide a useful indication of whether COCO inhibits reparative (intra-retinal) angiogenesis in this model.
9. Fig 7. Ethical approvals need to be stated in the methods for the human retina studies. What were the n numbers for these studies? (i.e. how many human retinas were used the Wb and IHC work?)
10. Fig 8. How was the transcriptomics data analysed? Details need to be added to the methods. Do the authors plan to make this data available for secondary analysis by other groups? The NAD⁺/NADH ratio data is shown in Fig 8G not 8F as stated in the main text.

Referee #2 (Remarks for Author):

The authors present an interesting paper dealing with the role of the COCO protein in retinal (neo)angiogenesis. Retinopathy of prematurity and wet age-related macular degeneration are important eye diseases with a prominent vascular component. Currently, anti-VEGF treatment is used in these patient cohorts. Due to the partially neuroprotective and developmental functions of VEGF, alternative treatments that do not affect survival of retinal cells are under active investigation in eye research.

Overall, the manuscript presents a solid piece of data that highlights a potential role of COCO in molecular medicine applications related to retinal angiogenesis. The paper is well written and follows a logical flow from in vitro to in vivo and includes also mechanistic studies. I think the paper could profit from a better characterization of endogenous COCO expression in the retina and more background information on biotechnological production of the recombinant COCO protein.

Specific points:

1. As the COCO protein was applied in vitro and in vivo, it is important to document how the protein was produced (in which cellular system), whether it was purified and otherwise functionally tested. If the recombinant COCO protein was available commercially, then the authors should state this in the methods part.

2. Endogenous COCO expression was shown in Figure 7. I think that an additional (antibody-independent) detection method should be used. The authors could perform in situ hybridization for example using RNAscope that allows a very sensitive and versatile detection together with potential other markers. As it stands, it is not fully clear which cells express COCO in the retina.

3. Figure 8 transcriptomics: It could be helpful to do a more sophisticated bioinformatic analysis related to molecular pathways. The ATP/mitochondrial story on the mechanism may be hidden in the gene list (Fig. 8A).

Referee #3 (Remarks for Author):

In this manuscript, the authors investigated COCO and its effects on angiogenesis in mouse eyes. They demonstrated that COCO inhibits sprouting, migration, wound closure and cellular proliferation in cultured HUVEC. They also found that the sprouting area, was decreased in COCO-treated choroidal explants. In vivo experiments, they showed that Intravitreal injections of COCO inhibited retinal vascularization in models of retinopathy of prematurity (OIR) and AMD (Laser induced CNV). For mechanism study, in cultured HUVEC, they showed that COCO inhibits glucose uptake and ATP production. In addition, HUVECs treated with COCO displayed a decrease of the NAD⁺/NADH ratio and increase of ROS. The authors concluded that COCO is an inhibitor of retinal and choroidal angiogenesis, mechanistically, in part by modulating ATP production in ECs.

It is a well-designed and very interesting study. The manuscript is well written as well.

Major concerns:

1. All of their in vitro studies were performed on HUVEC. It would be important to show the anti-angiogenic effects of COCO on HREC (human retinal endothelial cells). At least repeating some of the critical experiments in HREC.

2. For mechanism study, it might provide more information if assessing the energy metabolism using the Seahorse Flux Analyzer.

Minor concerns:

In Figure-1. "COCO inhibits sprouting angiogenesis. (A) Representative images of HUVECs sprouting in a fibrin gel in the presence or absence of VEGF and/or COCO...(C) Quantification of tube surface area of micrographs shown in (A)". Where can we find the "VEGF"?

In Discussion, "...One of the more surprising findings of this study was the observation that COCO affects energy metabolism in ECs, which is crucial for the angiogenic process (Eelen et al, 2018;

Treps et al, 2016)..." Those two references are review articles, and where is the "COCO" mentioned?

Referee #1 (Remarks for Author):

The paper by Popovic and colleagues examines the involvement of COCO/DAND5 in developmental and pathological angiogenesis in the eye. Using HUVECs and a choroidal explant model, the authors show that COCO inhibits sprouting, migration and proliferation of cultured endothelial cells by altering their metabolic and redox status. These findings are further supported by a range of in vivo studies demonstrating that intravitreal injection of COCO inhibits retinal vascularisation during developmental and pathological angiogenesis in mouse models of retinopathy of prematurity and wet AMD. The paper is well written and interesting, but the conclusions could be strengthened in some areas.

1. The observation that COCO modulates ATP production, glucose uptake and redox signalling in HUVECs is interesting. However, it remains unclear if these changes result from inhibition of Wnt, BMP or TGFbeta signalling or are a direct consequence of its uptake into the cells. Addressing this issue would help to better define mechanistically how COCO modulates the angiogenic process and greatly improve the overall conclusions of the manuscript.

Canonically, COCO, as a Dan family member, has been described to mediate its effects by modulating TGFbeta, Wnt and BMP signaling. All three of these signaling pathways have shown to have extensive effects on cellular respiration and metabolism. Depending on the context, TGFbeta stimulation can modulate mitochondrial respiration (Scientific Reports volume 7, Article number: 12486 (2017)), glycolysis (J Biol Chem . 2015 Oct 16;290(42):25427-38.) and ROS generation (Am J Physiol Renal Physiol. 2013 Nov 15; 305(10): F1477–F1490). Similarly, Wnt-driven Warburg metabolism characterized by increased reliance on glycolysis for energy production has been reported in several cell lines (EMBO J. 2014 Jul 1; 33(13): 1454–1473; Cell Metabolism, Volume 17, Issue 5, 7 May 2013, Pages 745-755). Given that we report a significant inhibition of TGFbeta signaling and BMP signaling in COCO-treated HUVECs (Figure 8), we anticipate that COCO modulates the energy metabolism of endothelial cells at least in part through the modulation of these pathways. However, given that COCO also appears to be

internalized in mitochondria shortly after stimulation (Fig. 7-D), we cannot rule out a direct effect of internalized COCO; the rapid effects of COCO on energy metabolism (Fig-9) similarly suggest that COCO could have a more direct effect unrelated to its quenching of TGFbeta and BMP ligands. It is therefore likely that the effects of COCO on cellular metabolism arise from a complex combination of its quenching activity combined with a more direct effect related to its internalization. However, while we now show Seahorse studies (Fig. 9A), which confirm that COCO decreases ECAR in endothelial cells, without performing proteomics studies combined with genetic loss-of-function experiments, which would be beyond the scope of this study, it will be difficult to dissociate these 2 activities of COCO. We have highlighted these issues in the discussion (pages 11-12).

2. The in vitro data shows that COCO inhibits angiogenesis through a mechanism that is independent of VEGF signalling. Therefore, COCO should have an additive effect with Flt1Fc in the in vivo models of retinal and choroidal angiogenesis. From a therapeutic perspective, this would seem important to confirm using at least one of the in vivo models.

In vitro, we now show that treatment with a sub-optimal concentration of COCO (50ng/mL) can potentiate the anti-angiogenic effects of Flt1Fc in a sprouting experiment (Supp Figure 3). However, no such potentiation was observed at higher COCO concentrations (75-300 ng/ml COCO), suggesting that the range in which COCO potentializes the effects of anti-VEGF is relatively narrow. In vivo, co-injections of COCO and FLT1Fc at 50ng/ml did not significantly potentiate the effects of Flt1Fc (Supp Fig. 3). At this dosage, the anti-angiogenic effects of COCO still appeared to be important, which may partially explain why an additive effect was not observed. It is possible that the concentration window required to observe an additive effect is relatively narrow both in vitro and in vivo. An in-depth characterization of the therapeutic range of COCO in the presence or absence of VEGF inhibitors is the focus of another study.

3. It remains unclear from the paper whether COCO plays any endogenous role in modulating developmental or pathological angiogenesis in the eye. All of the experiments involve intravitreally injecting the recombinant protein.

The current manuscript focuses on characterizing the consequences of exogenous COCO administration on retinal and choroidal angiogenesis. We are currently evaluating the effects of endogenous COCO on retinal vascular development in *cer12*^{-/-} mice obtained from Dr José A Belo. While no significant differences in the retinal vasculature have been observed between control and *cer12*^{-/-} animals at P5, we are currently characterizing a potential defect in diving vessels at P12. Functional studies involving ERG will also be performed to address the role of endogenous COCO on visual function. As we would like to include these data in an upcoming manuscript, which will focus on the characterization of endogenous COCO during vascular and retinal development as well as in pathological models, we chose not to include them in the current manuscript, to focus on the pre-clinical potential of exogenous COCO delivery in ocular angiogenesis.

4. There appears to be a major discrepancy between the concentrations of COCO used in the in vitro and in vivo experiments. For the in vitro experiments, concentrations up to 100ng/ml were used. For the mouse studies, it seems that 100ng was intravitreally injected into the mouse eye, which has a vitreal volume of around 5ul. It would be helpful if the authors could clarify the final vitreal concentration of COCO used, and if they have observed effects in vivo at more physiologically relevant protein concentrations.

We have clarified the concentrations used in the text. The concentrations described in the manuscript represent the final ocular concentrations of COCO in the vitreous, considering the volume of mouse vitreous (5ul) (P.14) (Vision Res. 2004;44(16):1857-67. doi: 10.1016/j.visres.2004.03.011.)

Minor Comments:

1. P15 - How was the normality of the data tested prior to applying parametric statistics?

Normality and homogeneity were shown with Shapiro-Wilk test

2. Fig 1E - The wound closure image shown for COCO appears to suggest that it is stimulating cell migration rather than inhibiting it.

We apologise, this is our mistake; the panels in this figure 1G were inverted between control and COCO; we have corrected it.

3. How was the apoptosis data in Fig 1I,J statistically analysed?

Error bars were added to the figure. Groups were compared by using Mann-Whitney test, but no statistical differences were noted.

4. In Fig 2B it appears that COCO is causing considerable inflammatory cell infiltration into the retina. It would be helpful if the authors could comment on this.

Inflammatory cell infiltration is sometimes observed following intravitreal injections. While we did seldom observe inflammatory cell infiltration following intravitreal injections, we did not observe that this was more prominent in the COCO group. We have replaced a panel in Fig. 2A to better reflect this.

5. In Figs 2C and 5F, were the effects of COCO significantly greater than those of Flt1Fc as suggested in the main manuscript text? If not, the text should be adjusted accordingly.

While there was a tendency for a greater effect of COCO compared to Flt1Fc on developmental retinal angiogenesis, this effect was not statistically significant. We have corrected this in the text.

6. Data in Figs 2D and E would benefit from quantification to enable firm conclusions to be drawn.

We have conducted quantifications for these effects (staining overlap), which are now shown in figure 2, and confirm that there are no significant differences in collagen IV deposition and pericyte coverage between groups.

7. In Fig 2F its unclear how changes in photoreceptor morphology could be determined, as suggested in the main text. Further clarification is needed here.

While the number of photoreceptor nuclei were similar and no pyknotic nuclei were observed between control and COCO-injected eyes, our stainings do not allow us to draw conclusions for photoreceptor morphology. As such, we have removed this statement from the manuscript.

8. Fig 5B - in addition to reporting the neovascular area in OIR experiments, it would be normal practice to also present data on avascular areas. This would provide a useful indication of whether COCO inhibits reparative (intra-retinal) angiogenesis in this model.

We have now provided quantification of the avascular areas in figure 5D.

9. Fig 7. Ethical approvals need to be stated in the methods for the human retina studies. What were the n numbers for these studies? (i.e. how many human retinas were used the Wb and IHC work?)

Ethical approval was specified in the Methods section; immunostaining in Fig. 7C was performed on one human eye, while the WB were performed on human cells on three different occasions (Fig. 7A,B). This was specified in the figure legend.

10. Fig 8. How was the transcriptomics data analysed? Details need to be added to the methods. Do the authors plan to make this data available for secondary analysis by other groups? The NAD⁺/NADH ratio data is shown in Fig 8G not 8F as stated in the main text.

We have now provided new analysis in figure 8, and included GSEA pathway analysis. Furthermore, we have increased the number of n for these studies, which allowed us to identify more genes significantly altered in response to COCO treatments. Details on data analysis were added in the Methods section. All raw data files of the transcriptomics data have been deposited in the Gene Expression Omnibus (GEO) database (accession # GSE160099). The following secure token has been created to allow review of record GSE160099 while it remains in private status: ivmbuyimxpwpsh

The NAD⁺/NADH ratio is now included in Figure-9, since we have separated the transcriptomics data from the metabolic analyses. This was changed in the results section.

Referee #2 (Remarks for Author):

In this manuscript, the authors investigated COCO and its effects on angiogenesis in mouse eyes. They demonstrated that COCO inhibits sprouting, migration, wound closure and cellular proliferation in cultured HUVEC. They also found that the sprouting area, was decreased in COCO-treated choroidal explants. In vivo experiments, they showed that Intravitreal injections of COCO inhibited retinal vascularization in models of retinopathy of prematurity (OIR) and AMD (Laser induced CNV). For mechanism study, in cultured HUVEC, they showed that COCO inhibits glucose uptake and ATP production. In addition, HUVECs treated with COCO displayed a decrease of the NAD⁺/NADH ratio and increase of ROS. The authors concluded that COCO is an inhibitor of retinal and choroidal angiogenesis, mechanistically, in part by modulating ATP production in ECs.

It is a well-designed and very interesting study. The manuscript is well written as well.

Major concerns:

1. All of their in vitro studies were performed on HUVEC. It would be important to show the anti-angiogenic effects of COCO on HREC (human retinal endothelial cells). At least repeating some of the critical experiments in HREC.

We have performed sprouting experiments on Human Microvascular Retinal Endothelial Cells, which confirm that the effects of COCO are consistent in retinal microvascular endothelial cells. These data are shown in figure-1B,E

2.For mechanism study, it might provide more information if assessing the energy metabolism using the Seahorse Flux Analyzer.

We have performed Seahorse analysis, which show that COCO treatment lowers ECAR, an indicator of glycolysis, in endothelial cells. These data are shown in figure-9A.

Minor concerns:

In Figure-1. "COCO inhibits sprouting angiogenesis. (A) Representative images of HUVECs sprouting in a fibrin gel in the presence or absence of VEGF and/or COCO...(C) Quantification of tube surface area of micrographs shown in (A)". Where can we find the "VEGF"?

For sprouting assays, cells were cultured in complete EBM2 medium supplemented with 25 ng/ml VEGF, with or without increasing concentrations of COCO. VEGF is present in all conditions, and this was corrected in the figure legend and specified in the methods.

In Discussion, "...One of the more surprising findings of this study was the observation that COCO affects energy metabolism in ECs, which is crucial for the angiogenic process (Eelen et al, 2018; Treps et al, 2016)..." Those two references are review articles, and where is the "COCO" mentioned?

Our apologies; we realize that the phrasing created confusion. These reviews were cited to highlight the importance of metabolism on the angiogenic process, and not meant to show that COCO had been described to play a role in endothelial metabolism, which, to our knowledge has not been shown previously. We have rephrased this statement and cited original studies to clarify this statement. It now reads: "One of the more surprising findings of this study was the observation that COCO affects energy metabolism in ECs. Several studies have highlighted the importance of metabolic regulation in the endothelium and shown the critical importance of metabolic and glycolytic pathways in driving the angiogenic process (Eelen et al, 2018; Treps et al, 2016; De Bock et al, 2013; Cantelmo et al, 2016)."

Referee #3 (Remarks for Author):

The authors present an interesting paper dealing with the role of the COCO protein in retinal (neo)angiogenesis. Retinopathy of prematurity and wet age-related macular degeneration are important eye diseases with a prominent vascular component. Currently, anti-VEGF treatment is used in these patient cohorts. Due to the partially neuroprotective and developmental functions of VEGF, alternative treatments that do not affect survival of retinal cells are under active investigation in eye research.

Overall, the manuscript presents a solid piece of data that highlights a potential role of COCO in molecular medicine applications related to retinal angiogenesis. The paper is well written and follows a logical flow from in vitro to in vivo and includes also mechanistic studies. I think the paper could profit from a better characterization of endogenous COCO expression in the retina and more background information on biotechnological production of the recombinant COCO protein.

Specific points:

1. As the COCO protein was applied in vitro and in vivo, it is important to document how the protein was produced (in which cellular system), whether it was purified and otherwise functionally tested. If the recombinant COCO protein was available commercially, then the authors should state this in the methods part.

The protein used was obtained commercially from R&D systems and produced in E. coli. The details and catalog number (3047-cc) were provided in the Methods section

2. Endogenous COCO expression was shown in Figure 7. I think that an additional (antibody-independent) detection method should be used. The authors could perform in situ hybridization for example using RNAscope that allows a very sensitive and versatile detection together with potential other markers. As it stands, it is not fully clear which cells express COCO in the retina.

We have performed RNAscope immunostaining on mouse retinas, which confirm COCO expression in the neural retina at P5. Panels are shown in figure 7D,E and show that COCO is detected throughout the neural retina and is present in the photoreceptor nuclear layer, as shown by co-localization with and visual-arrestin immunostaining. Cells in the inner nuclear layer, composed of horizontal, bipolar and amacrine cells, as well as in the ganglion cell layer also strongly expressed COCO.

3. Figure 8 transcriptomics: It could be helpful to do a more sophisticated bioinformatic analysis related to molecular pathways. The ATP/mitochondrial story on the mechanism may be hidden in the gene list (Fig. 8A).

We thank the reviewers for noticing the lack of details in the transcriptomic analysis. In order to better reply reviewer's comments, we have analyzed more independent samples and updated figure 8. Besides adding more details in the methods section for the data analysis, we added more insights towards the dysregulation of TGFb and BMP signaling and oxidative stress pathways by showing Gene Set Enrichment Analysis (GSEA). We also showed the list of top-altered genes ($\text{Log}_2\text{Fc} \geq \pm 1$; $\text{Pvalue} \leq 0.05$) of HUVECs treated with COCO, which showed expression changes in several mitochondrial genes.

18th Nov 2020

Dear Dr. Larrivee,

Thank you for the submission of your revised manuscript to EMBO Molecular Medicine. I am pleased to inform you that we will be able to accept your manuscript pending the following final amendments:

1) In the main manuscript file, please do the following:

- Correct/answer the track changes suggested by our data editors by working from the attached/uploaded document

***** Reviewer's comments *****

Referee #1 (Remarks for Author):

The authors have adequately addressed my previous comments.

Referee #2 (Remarks for Author):

The authors have done a very good job in addressing my previous concerns with novel experiments. Especially adding novel in situ hybridization data (Figure 7) and including n=4 independent samples per group in transcriptomic analyses of HUVECs (Figure 8) strengthen the findings and put a clearer focus on COCO expression in the mouse retina and COCO-mediated signaling, respectively. I have no further reservations.

Referee #3 (Comments on Novelty/Model System for Author):

The authors have addressed my concerns. The manuscript is improved and is a substantial contribution to the field.

Referee #3 (Remarks for Author):

The authors have addressed my concerns. The manuscript is improved and is a substantial contribution to the field.

The authors performed the requested editorial changes.

24th Dec 2020

Dear Dr. Larrivee,

We are pleased to inform you that your manuscript is accepted for publication.

Corresponding Author Name: Bruno Larrivée

Manuscript Number: EMM-2020-12005-V2